# Deep Learning meets Nonparametric Regression: Are Weight-Decayed DNNs Locally Adaptive?

## Abstract

We study the theory of neural network (NN) from the lens of classical nonparametric regression problems with a focus on NN's ability to *adaptively* estimate functions with *heterogeneous smoothness* — a property of functions in Besov or Bounded Variation (BV) classes. Existing work on this problem requires tuning the NN architecture based on the function spaces and sample sizes. We consider a "Parallel NN" variant of deep ReLU networks and show that the standard weight decay is equivalent to promoting the $\ell_p$-sparsity ($0 < p < 1$) of the coefficient vector of an end-to-end learned function bases, i.e., a dictionary. Using this equivalence, we further establish that by tuning only the weight decay, such Parallel NN achieves an estimation error arbitrarily close to the minimax rates for both the Besov and BV classes. Notably, it gets exponentially closer to minimax optimal as the NN gets deeper. Our research sheds new lights on why depth matters and how NNs are more powerful than kernel methods.

## 1 Introduction

*Why* do deep neural networks (DNNs) *work* better? They are universal function approximators [6], but so are splines and kernels. They learn data-driven representations, but so are the shallower and linear counterparts such as matrix factorization. There is surprisingly little theoretical understanding on why DNNs are superior to these classical alternatives.

In this paper, we study DNNs in nonparametric regression problems — a classical branch of statistical theory and methods with more than half a century of associated literature [25, 7, 46, 10, 23, 37, 33]. Nonparametric regression addresses the following fundamental problem:

- Let $y_i = f(x_i) +$ Noise for $i = 1, ..., n$. How can we estimate a function $f$ using data points $(x_1, y_1), ..., (x_n, y_n)$ in conjunction with the knowledge that $f$ belongs to a function class $\mathcal{F}$?

Function class $\mathcal{F}$ typically imposes only weak regularity assumptions such as smoothness, which makes nonparametric regression widely applicable to real-life applications under weak assumptions.

**Local adaptivity.** A subset of nonparametric regression techniques were shown to have the property of *local adaptivity* [24] in both theory and practice. These include wavelet smoothing [10], locally adaptive regression splines [24], trend filtering [40, 47] and adaptive local polynomials [2, 3]. We say a nonparametric regression technique is *locally adaptive* if it can cater to local differences in smoothness, hence allowing more accurate estimation of functions with varying smoothness and abrupt changes.

In light of such a distinction, it is natural to consider the following question.

> Are NNs *locally adaptive*, i.e., optimal in learning functions with heterogeneous smoothness?

Submitted to 36th Conference on Neural Information Processing Systems (NeurIPS 2022). Do not distribute.

This is a timely question to ask, partly because the bulk of recent theory of NN leverages its asymptotic Reproducing Kernel Hilbert Space (RKHS) in the overparameterized regime [21, 5, 1]. RKHS-based approaches, e.g., kernel ridge regression with any fixed kernels are *suboptimal* in estimating functions with heterogeneous smoothness [9]. Therefore, existing deep learning theory based on RKHS does not satisfactorily explain the advantages of neural networks over kernel methods.

We build upon the recent work of Suzuki [39] and Parhi and Nowak [29] who provided encouraging first answers to the question above about the local adaptivity of NNs. Specifically, Parhi and Nowak [29, Theorem 8] showed that a two-layer *truncated power function* activated neural network with a non-standard regularization is equivalent to the *locally adaptive regression splines* (LARS) [24]. This connection implies that such non-standard NNs achieve the minimax rate for the (higher order) bounded variation (BV) classes. We provide a detailed discussion about this work in Section B. Suzuki [39] showed that multilayer ReLU DNNs can achieve minimax rate for the Besov class, but requires the width, depth and an artificially imposed sparsity-level of the DNN weights to be carefully calibrated according to parameters of the Besov class, thus is quite different from how DNNs are typically trained in practice.

In this paper, we aim at addressing the same *locally adaptivity* question for a more commonly used neural network with standard weight decayed training.

**Parallel neural networks.** We restrict our attention on a special network architecture called *parallel neural network* [18, 15] which learns an ensemble of subnetworks — each being a multilayer ReLU DNNs. Parallel NNs have been shown to be more well-behaved both theoretically [18, 51, 16, 15, 14] and empirically [50, 44]. Moreover, the idea of parallel NNs was used in many successful NN architectures such as SqueezeNet, ResNext and Inception (see [15] and the references therein).

**Weight decay.** Weight decay is a common method in deep learning to reduce overfitting. Empirically, the regularizer is not necessarily explicit. Many tricks in deep learning, including early stopping [48], quantization [20], and dropout [45] have similar effect as weight decay. In this paper, we make no assumption on the training method thus there is no (implicit) regularizers apart from weight decay.

**Summary of results.** Our main contributions are:

1. We prove that the (standard) weight decay in training an $L$-layer *parallel* ReLU-activated neural network is equivalent to a sparse $\ell_p$ penalty term (where $p = 2/L$) on the linear coefficients of a learned representation.

2. We show that neural networks can approximate B-spline basis functions of any order without the need of choosing the order parameter manually. In other words, neural networks can adapt to functions of different order of smoothness, and even functions with different smoothness in different regions in their domain.

3. We show that the estimation error of weight decayed parallel ReLU neural network decreases polynomially with the number of samples up to a constant error for estimating functions with heterogeneous smoothness in the both BV and Besov classes, and the exponential term in the error rate is close to the minimax rate. Notably, the method requires tuning only the weight decay parameter.

4. We find that deeper models achieve closer to the optimal error rate. This result helps explain why deep neural networks can achieve better performance than shallow ones empirically.

The above results separate NNs with any linear methods such as kernel ridge regression. To the best of our knowledge, we are the first to demonstrate that standard techniques ("weight decay" and ReLU activation) suffice for DNNs in achieving the optimal rates for estimating BV and Besov functions.

## 2 Preliminary

### 2.1 Notation and Problem Setup.

We denote regular font letters as scalars, bold lower case letters as vectors and bold upper case letters as matrices. $a \lesssim b$ means $a \leq Cb$ for some constant $C$ that does not depend on $a$ or $b$, and $a \asymp b$ denotes $a \lesssim b$ and $b \lesssim a$. See Table 1 for the full list of symbols used.

Table 1: Symbols used in this paper

| symbol | Meaning | | |
|---|---|---|---|
| $a/\boldsymbol{a}/\mathbf{A}$ | scalars / vectors / matrices. | $[a,b]$ | $\{x \in \mathbb{R} : a \leq x \leq b\}$ |
| $B_{p,q}^{\alpha}$ | Besov space. | $[n]$ | $\{x \in \mathbb{N} : 1 \leq x \leq n\}$. |
| $\|\cdot\|_{B_{p,q}^{\alpha}}$ | Besov quasi-norm . | $\|\cdot\|_F$ | Frobenius norm. |
| $\|\cdot\|_{B_{p,q}^{\alpha}}$ | Besov norm. | $\|\cdot\|_p$ | $\ell_p$-norm. |
| $M_m(\cdot)$ | $m^{th}$ order Cardinal B-spline bases. | $d$ | Dimension of input. |
| $M_{m,k,\boldsymbol{s}}(\cdot)$ | $m^{th}$ order Cardinal B-spline basis function of resolution $k$ at position $\boldsymbol{s}$. | $M$ $L$ | # subnetworks in a parallel NN. # layers in a (parallel) NN. |
| | | $w$ | Width of a subnetwork. |
| $\sigma(\cdot)$ | ReLU activation function. | $n$ | # samples. |
| $\mathbf{W}_j^{(\ell)}, \boldsymbol{b}_j^{(\ell)}$ | Weight and bias in the $\ell$-th layer in the $j$-th subnetwork. | $\mathbb{R}, \mathbb{Z}, \mathbb{N}$ | Set of real numbers, integers, and nonnegative integers. |

Let $f_0$ be the target function to be estimated. The training dataset is $\mathcal{D}_n := \{(\boldsymbol{x}_i, y_i), y_i = f_0(\boldsymbol{x}_i) + \epsilon_i, i \in [n]\}$, where $x_i$ are fixed and $\epsilon_i$ are zero-mean, independent Gaussian noises with variance $\sigma^2$. In the following discussion, we assume $\boldsymbol{x}_i \in [0,1]^d$, $f_0(x_i) \in [-1,1], \forall i$.

We will be comparing estimators under the mean square error (MSE), defined as

$$\text{MSE}(\hat{f}) := \mathbb{E}_{\mathcal{D}_n} \frac{1}{n} \sum_{i=1}^{n} (\hat{f}(\boldsymbol{x}_i) - f_0(\boldsymbol{x}_i))^2.$$

The optimal worst-case MSE is described by $R(\mathcal{F}) := \min_{\hat{f}} \max_{f_0 \in \mathcal{F}} \text{MSE}(\hat{f})$, we say that $\hat{f}$ is optimal if $\text{MSE}(\hat{f}) \asymp R(\mathcal{F})$. The empirical (square error) loss is defined as $\hat{L}(\hat{f}) := \frac{1}{n} \sum_{i=1}^{n} (\hat{f}(\boldsymbol{x}_i) - y_i)^2$. The corresponding population loss is $L(\hat{f}) := \mathbb{E}[\frac{1}{n} \sum_{i=1}^{n} (\hat{f}(\boldsymbol{x}_i) - y_i')^2 | \hat{f}]$ where $y_i'$ are new data points. It is clear that $\mathbb{E}[L(\hat{f})] = \text{MSE}[\hat{f}] + \sigma^2$.

## 2.2 Besov Spaces and Bound Variation Space

**Besov space**, denoted as $B_{p,q}^{\alpha}$, is a flexible function class parameterized by $\alpha, p, q$ whose definition is deferred to Section C.1. Here $\alpha \geq 0$ determines the smoothness of functions, $1 \leq p \leq \infty$ determines the averaging (quasi-)norm over locations, $1 \leq q \leq \infty$ determines the averaging (quasi-)norm over scale which plays a relatively minor role. Smaller $p$ is more forgiving to inhomogeneity and loosely speaking, when the function domain is bounded, smaller $p$ induces a larger function space. On the other hand, it is easy to see from definition that $B_{p,q}^{\alpha} \subset B_{p,q'}^{\alpha}$, if $q < q'$. Without loss of generalizability, in the following discussion we will only focus on $B_{p,\infty}^{\alpha}$.

When $p = 1$, the Besov space allows higher inhomogeneity, and it is more general than the Sobolev or Hölder space.

**Bounded variation (BV) space** is a more interpretable class of functions with spatially heterogeneous smoothness [10]. It is defined through the total variation (TV) of a function. For $(m+1)$th differentiable function $f : [0,1] \to \mathbb{R}$, the $m$th order total variation is defined as $TV^{(m)}(f) := TV(f^{(m+1)}) = \int_{[0,1]} |f^{(m+1)}(x)| dx$, and the corresponding $m$th order Bounded Variation class $BV(m) := \{f : TV(f^{(m)}) < \infty\}$. The more general definition is given in Section C.2. Bounded variation class is tightly connected to Besov classes. Specifically [8]:

$$B_{1,1}^{m+1} \subset BV(m) \subset B_{1,\infty}^{m+1} \tag{1}$$

This allows the results derived for the Besov space to be easily applied to BV space.

**Minimax MSE** It is well known that minimax rate for Besov and 1D BV classes are $O(n^{-\frac{2\alpha}{2\alpha+d}})$ and $O(n^{-(2m+2)/(2m+3)})$ respectively . The minimax rate for *linear estimators* in 1D BV classes is known to be $O(n^{-(2m+1)/(2m+2)})$ [24, 10].

## 3 Main Results: Parallel ReLU DNNs

Consider a parallel neural network containing $M$ multi layer perceptrons (MLP) with ReLU activation functions called *subnetworks*. Each subnetwork has width $w$ and depth $L$. The input is fed to

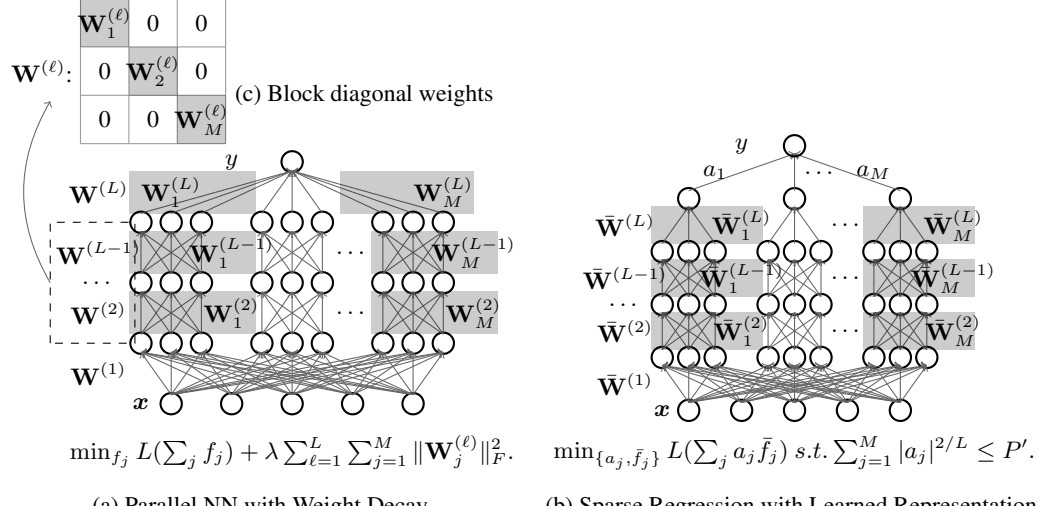

Figure 1: Parallel neural network and the equivalent sparse regression model we discovered.

all the subnetworks, and the output of the parallel NN is the summation of the output of each sub-network. The architecture of a parallel neural network is shown in Figure 1a. This parallel neural network is equivalent to a vanilla neural network with block diagonal weights in all but the first and the last layers (Figure 1(c)). Let $\mathbf{W}_j^{(\ell)}$ and $\boldsymbol{b}_j^{(\ell)}$ denote the weight and bias in the $\ell$-th layer in the $j$-th subnetwork respectively. Training this model with weight decay returns:

$$\underset{\{\mathbf{W}_j^{(\ell)},\boldsymbol{b}_j^{(\ell)}\}}{\arg\min}\ \hat{L}(f) + \lambda \sum_{j=1}^{M}\sum_{\ell=1}^{L} \big\|\mathbf{W}_j^{(\ell)}\big\|_F^2, \qquad (2)$$

where $f(x) = \sum_{j=1}^{M} f_k(x)$ denotes the parallel neural network, $f_j(\cdot)$ denotes the $j$-th subnetwork, and $\lambda > 0$ is a fixed scaling factor.

**Theorem 1.** *For any fixed $\alpha - d/p > 1, q \geq 1, L \geq 3$, for any $f_0 \in B_{p,q}^\alpha$, given an $L$-layer parallel neural network satisfying*

- *The width of each subnetwork is fixed and large enough: $w \gtrsim d$. See Theorem 9 for the detail.*

- *The number of subnetworks is large enough: $M \gtrsim m^d n^{\frac{1-2/L}{2\alpha/d+1-2/(pL)}}$ where $m = \lceil \alpha - 1 \rceil$.*

*With proper choice of the parameter of weight decay $\lambda$, the solution $\hat{f}$ parameterized by (2) satisfies*

$$\mathrm{MSE}(\hat{f}) = \tilde{O}\big(n^{-\frac{2\alpha/d(1-2/L)}{2\alpha/d+1-2/(pL)}}\big) + Const. \qquad (3)$$

*where $\tilde{O}$ shows the scale up to a logarithmic factor, and the trailing constant term decreases exponentially with $L$.*

We explain the proof idea in the next section, but defer the extended form of the theorem and the full proof to Section F. Before that, we comment on a few interesting aspects of the result.

**Near optimal rates and the effect of depth.** The first term in the MSE bound is the estimation error and the second term is (part of) the approximation error of this NN. Recall that the minimax rate of a Besov class is $O(n^{-\frac{2\alpha}{2\alpha+d}})$ thus as the depth parameter $L$ increases it can get arbitrarily close to the minimax rate. The constant term would be a negligible if we choose $L \gtrsim \log n$.

**Corollary 2.** *Under the conditions of Theorem 1, for any $f_0 \in B_{p,q}^\alpha$, there is a numerical constant $C$ such that when we choose $C \log n \leq L \leq 100 C \log n$,*

$$\mathrm{MSE}(\hat{f}) = \tilde{O}(n^{-\frac{2\alpha}{2\alpha+d}(1-o(1))}),$$

*where $\tilde{O}$ hides only logarithmic factors and the $o(1)$ factor in the exponent is $O(1/\log(n))$.*

This result says that deeper parallel neural networks achieves lower error and gets closer to the statistical limit.

**Overparameterization and sparsity.** We also note that the result does not depend on $M$ as long as $M$ is large enough. This means that the neural network can be arbitrarily overparameterized while not overfitting. The underlying reason is *sparsity*. As it will become clearer in the proof sketch, weight decayed training of a parallel L-layer ReLU NNs is equivalent to a sparse regression problem with an $\ell_p$ penalty assigned to the coefficient vector of a learned dictionary. Here $p = 2/L$ which promotes even sparser solutions than an $\ell_1$ penalty.

**No architecture tuning.** For any fixed $L$, the required architecture of the model does not depend on the dataset or the target function $(n, \alpha)$ expect the number of subnetworks $M$, for which the only requirement is being large enough. As a result, one can design a model using a large guess on $M$, and achieve the claimed near-optimal error rate by only tuning the weight decay parameter.

**Bounded variation classes.** Thanks to the Besov space embedding of the BV class (1), our theorem also implies the result for the BV class in $1D$.

**Corollary 3.** *If the target function is in bounded variation class $f_0 \in BV(m)$, For any fixed $L \geq 3$, for a neural network satisfying the requirements in Theorem 1 with $d = 1$ and with proper choice of the parameter of weight decay $\lambda$, the NN $\hat{f}$ parameterized by (5) satisfies*

$$\mathrm{MSE}(\hat{f}) = \tilde{O}(n^{-\frac{(2m+2)(1-2/L)}{2m+3-2/L}}) + Const.$$

*where $\tilde{O}$ shows the scale up to a logarithmic factor, and the trailing constant term decreases exponentially with L.*

It is known that any linear estimators such as kernel smoothing and smoothing splines cannot have an error lower than $O(n^{-(2m+1)/(2m+2)})$ for $BV(m)$ [10]. This partly explains the advantage of DNNs over kernels.

**Representation learning and adaptivity.** The results also shed a light on the role of representation learning in DNN's ability to adapt. Specifically, different from the two-layer NN in [29], which achieves the minimax rate of $BV(m)$ by choosing appropriate activation functions using each $m$, each subnetwork of a parallel NN can learn to approximate the spline basis of an arbitrary order, which means that if we choose $L$ to be sufficiently large, such Parallel NN with optimally tuned $\lambda$ is simultaneously near optimal for $m = 1, 2, 3, \ldots$. In fact, even if different regions of the space has different *orders* of smoothness, the paralle NN will still be able to learn appropriate basis functions in each local region. To the best of our knowledge, this is a property that none of the classical nonparametric regression methods possess.

**Synthesis v.s. analysis methods.** Our result could also inspire new ideas in estimator design. There are two families of methods in non-parametric estimation. One called *synthesis* framework which focuses on constructing appropriate basis functions to encode the contemplated structures and regress the data to such basis, e.g., wavelets [10]. The other is called *analysis* framework which uses analysis regularization on the data directly (see, e.g., RKHS methods [37] or trend filtering [40]). It appears to us that parallel NN is doing both simultaneously. It has a parametric family capable to synthesizing an $O(n)$ subset of an exponentially large family of basis, then *implicitly* use sparsity-inducing analysis regularization to select the relevant basis functions. In this way the estimator does not actually have to explicitly represent that exponentially large set of basis functions, thus computationally more efficient.

# 4 Proof Overview

We start by first proving that a parallel neural network trained with weight decay is equivalent to an $\ell_p$-sparse regression problem with representation learning (Section 4.1); which helps decompose its MSE into an estimation error and approxmation error. Then we bound the two terms in Section 4.2 and Section 4.3 respectively.

## 4.1 Equivalence to $\ell_p$ Sparse Regression with a Learned Feature Representation

It is widely known that ReLU function is 1-homogeneous: $\sigma(ax) = a\sigma(x), \forall a \geq 0, x \in \mathbb{R}$. In any consecutive two layers in a neural network (or a subnetwork), one can multiply the weight and bias

in one layer with a positive constant, and divide the weight in another layer with the same constant. The neural network after such transformation is equivalent to the original one:

$$\mathbf{W}^{(2)}\sigma(\mathbf{W}^{(1)}\boldsymbol{x} + \boldsymbol{b}^{(1)}) = \frac{1}{c}\mathbf{W}^{(2)}\sigma(c\mathbf{W}^{(1)}\boldsymbol{x} + c\boldsymbol{b}^{(1)}), \quad \forall c > 0, \boldsymbol{x}. \tag{4}$$

This property allows us to reformulate (2) to an $\ell_p$ sparsity constraint problem:

**Proposition 4.** *Fix the input dataset $\mathcal{D}_n$ and a constant $c_1 > 0$. There exists an one-to-one mapping between $\lambda > 0$ and $P' > 0$ such that (2) is equivalent to the following problem:*

$$\underset{\{\bar{\mathbf{W}}_j^{(\ell)}, \bar{\boldsymbol{b}}_j^{(\ell)}, a_j\}}{\arg\min} \hat{L}\Big(\sum_{j=1}^{M} a_j \bar{f}_j\Big) = \frac{1}{n}\sum_i (y_i - \bar{f}_{1:M}(\boldsymbol{x}_i)^T \boldsymbol{a})^2$$

$$s.t. \ \|\bar{\mathbf{W}}_j^{(1)}\|_F \le c_1\sqrt{d}, \forall j \in [M], \tag{5}$$

$$\|\bar{\mathbf{W}}_j^{(\ell)}\|_F \le c_1\sqrt{w}, \forall j \in [M], 2 \le \ell \le L, \quad \|\{a_j\}\|_{2/L}^{2/L} \le P'$$

*where $\bar{f}_j(\cdot)$ is a subnetwork with parameters $\bar{\mathbf{W}}_j^{(\ell)}, \bar{\boldsymbol{b}}_j^{(\ell)}$.*

This equivalent model is demonstated in Figure 1b. The proof can be found in Section D.1. The constraint $\|\bar{\mathbf{W}}_j^{(1)}\|_F \lesssim \sqrt{d}, \|\bar{\mathbf{W}}_j^{(\ell)}\|_F \lesssim \sqrt{w}, \forall \ell > 1$ is typical in deep learning for better numerical stability. The equivalent model in Proposition 4 is also a parallel neural network, but it appends one layer with parameters $\{a_k\}$ at the end of the neural network and the constraint on the Frobenius norm is converted to the $2/L$ norm on the factors $\{a_k\}$. Since $L \gg 2$ in a typical application, $2/L \ll 1$ and this constraint can enforce a sparser model than that in Section B.

There are two useful implications of Proposition 4. First, it gives an intuitive explanation on how a weight decayed Parallel NN works. Specifically, it can be viewed as a sparse linear regression with representation learning. Second, the conversion into the constrained form allows us to adapt generic statistical learning machinery (a self-bounding argument) from Suzuki [39, Proposition 4] for studying this constrained ERM problem.

The adaptation is nontrivial because (1) our regression problem has a *fixed design* (so data points are not iid); (2) there is an *unconstrained* subspace with no bounded metric entropy. Specifically, our Proposition 15 shows that the MSE of the regression problem can be bounded by

$$\text{MSE}(\hat{f}) = O\Big( \underbrace{\inf_{f \in \mathcal{F}} \text{MSE}(f)}_{\text{approximation error}} + \underbrace{\frac{\log\mathcal{N}(\mathcal{F}_\|, \delta, \|\cdot\|_\infty) + d(\mathcal{F}_\perp)}{n} + \delta}_{\text{estimation error}} \Big)$$

in which $\mathcal{F}$ decomposes into $\mathcal{F}_\| \times \mathcal{F}_\perp$, where $\mathcal{F}_\perp$ is an unconstrained subspace with finite dimension, and $\mathcal{F}_\|$ is a compact set in the orthogonal complement with a $\delta$-covering number of $\mathcal{N}(\mathcal{F}_\|, \delta, \|\cdot\|_\infty)$ in $\|\cdot\|_\infty$-norm. This decomposes MSE into an approximation errorand an estimation error. The novel analysis of these two represents the major technical contribution of this paper.

## 4.2 Estimation Error Analysis

The decomposition above reveals that to bound the estimation error, it suffices to compute the covering number of the constraint set in the sup-norm of the function it represents.

Previous results that bound the covering number of neural networks [49, 39] depends on the width of the neural networks explicitly, which cannot be applied when analysing a potentially infinitely wide neural network. In this section, we leverage the $\ell_p$-norm bounded coefficients to avoid the dependence in $M$ in the covering number bound.

**Theorem 5.** *The covering number of the model defined in (5) apart from the bias in the last layer satisfies*

$$\log\mathcal{N}(\mathcal{F}, \delta) \lesssim w^{2+2/(1-2/L)}L^2\sqrt{d}P'^{\frac{1}{1-2/L}}\delta^{-\frac{2/L}{1-2/L}}\log(wP'/\delta). \tag{6}$$

The proof can be found in Section D.2. It requires the following lemma:

**Lemma 6.** $\log \mathcal{N}(\mathcal{G}, \delta) \lesssim k \log(1/\delta)$ *for some finite $c_3$, and for any $g \in \mathcal{G}, |a| \leq 1$, we have $ag \in \mathcal{G}$. The covering number of $\mathcal{F} = \left\{ \sum_{i=1}^{M} a_i g_i \middle| g_i \in \mathcal{G}, \|a\|_p^p \leq P, 0 < p < 1 \right\}$ for any $P > 0$ satisfies*

$$\log \mathcal{N}(\mathcal{F}, \epsilon) \lesssim k P^{\frac{1}{1-p}} (\delta/c_3)^{-\frac{p}{1-p}} \log(c_3 P / \delta)$$

*up to a double logarithmic factor.*

See Section D.3 for the proof of Lemma 6. The covering number in Theorem 5 does not depend on the number of subnetworks $M$. In other words, it provides a bound of estimation error for an arbitrarily wide parallel neural network as long as the total Frobenius norm is bounded.

### 4.3 Approximation Error Analysis

The approximation error analysis involves two steps. In Section 4.3.1, we analyse how a subnetwork can approximate a B-spline basis. Then in Section 4.3.2 we show that a sparse linear combination of B-spline bases approximates Besov functions. Both add up to the total error in approximating Besov functions with a parallel neural network (Theorem 9).

#### 4.3.1 Approximation Error of B-spline Basis Function

As is shown in Section C.1, functions in Besov space can be alternatively represented in a sequence space via the coefficients of a cardinal B-spline basis. In this section we study the approximation ability of ReLU neural networks to B-spline basis function.

**Proposition 7.** *Let $M_{m,k,s}$ be the B-spline of order $m$ with scale $2^{-k}$ in each dimension and position $s \in \mathbb{R}^d$: $M_{m,k,s}(\boldsymbol{x}) := M_m(2^k(\boldsymbol{x} - \boldsymbol{s}))$, $M_m$ is defined in (11). There exists a parallel neural network that has the structure and satisfy the constraint in Proposition 4 for $d$-dimensional input and one output, containing $M = O(m^d)$ subnetworks, each of which has width $w = O(d)$ and depth $L = O(\log(c(m,d)/\epsilon))$ for some constant $w, c$ that depends only on $m$ and $d$, denoted as $\tilde{M}_m(\boldsymbol{x}), \boldsymbol{x} \in \mathbb{R}^d$, such that*

- $|\tilde{M}_{m,k,s}(\boldsymbol{x}) - M_{m,k,s}(\boldsymbol{x})| \leq \epsilon$, *if $0 \leq 2^k(x_i - s_i) \leq m + 1, \forall i \in [d]$,*

- $\tilde{M}_{m,k,s}(\boldsymbol{x}) = 0$, *otherwise.*

- *The weights in the last layer satisfy $\|a\|_{2/L}^{2/L} \lesssim 2^k m^d e^{2md/L}$.*

The proof can be found in Section E.1. Note that the product of the coefficients among all the layers are proportional to $2^k$, instead of $2^{km}$ when approximating truncated power basis functions. This is because the transformation from $M_m$ to $M_{m,k,s}$ only scales the domain of the function by $2^k$, while the codomain of the function is not changed. To apply the transformation to the neural network, one only need to scale weights in the first layer by $2^k$, which is equivalent to scaling the weights in each layer bt $2^{k/L}$ and adjusting the bias according.

#### 4.3.2 Approximation Error in Besov Space

With the results given in Section 4.3.1, we can estimate the approximation error of parallel ReLU neural networks to functions in Besov space.

**Proposition 8.** *Let $\alpha - d/p > 1, r > 0$. For any function in Besov space $f_0 \in B_{p,q}^\alpha$ and any positive integer $\bar{M}$, there is an $\bar{M}$-sparse approximation using B-spline basis of order $m$ satisfying $0 < \alpha < \min(m, m - 1 + 1/p)$: $\check{f}_{\bar{M}} = \sum_{i=1}^{\bar{M}} a_{k_i, s_i} M_{m, k_i, s_i}$ for any positive integer $\bar{M}$ such that the approximation error is bounded as $\|\check{f}_{\bar{M}} - f_0\|_r \lesssim \bar{M}^{-\alpha/d} \|f_0\|_{B_{p,q}^\alpha}$, and the coefficients satisfy*

$$\|\{2^{k_i} a_{k_i, s_i}\}_{k_i, s_i}\|_p \lesssim \|f_0\|_{B_{p,q}^\alpha}.$$

The proof can be found in Section E.2.

**Remark 1.** *The requirement in Proposition 8: $\alpha - d/p > 1$ is stronger than the condition typically found in approximation theorem $\alpha - d/p \geq 0$ [11], so-called "Boundary of continuity", or the condition in Suzuki [39] $\alpha > d(1/p - 1/r)_+$ . This is because although the functions in $B_{p,q}^\alpha$ when*

$0 \le \alpha - d/p < 1$ *can be approximated by B-spline basis, the sum of weighted coefficients may not converge. One simple example is the step function* $f_{step}(x) = \mathbf{1}(x \ge 0.5), f_{step} \in B^1_{1,\infty}$. *Although it can be decomposed using first order B-spline basis as in (10), the summation of the coefficients is infinite. Actually one only needs a ReLU neural network with one hidden layer and two neurons to approximate this function to arbitrary precision, but the weight need to go to infinity.*

**Theorem 9.** *Under the same condition as Proposition 8, for any positive integer $\bar{M}$, any function in Besov space $f_0 \in B^\alpha_{p,q}$ can be approximated by a parallel neural network with no less than $O(m^d \bar{M})$ number of subnetworks satisfying:*

      *1. Each subnetwork has width $w = O(d)$ and depth $L$.*

      *2. The weights in each layer satisfy $\|\bar{\mathbf{W}}^{(\ell)}_k\|_F \le O(\sqrt{w})$ except the first layer $\|\bar{\mathbf{W}}^{(1)}_k\|_F \le O(\sqrt{d})$,*

      *3. The scaling factors have bounded $2/L$-norm: $\|\{a_j\}\|^{2/L}_{2/L} \lesssim m^d e^{2md/L} \bar{M}^{1-2/(pL)}$.*

      *4. The approximation error is bounded by*
$$\|\tilde{f} - f_0\|_r \le (c_4 \bar{M}^{-\alpha/d} + c_5 e^{-c_6 L})\|f\|_{B^\alpha_{p,q}}$$
      *where $c_4, c_5, c_6$ are constants that depend only on $m, d$ and $p$.*

Here $\bar{M}$ is the number of "active" subnetworks, which is not to be confused with the number of subnetworks at initialization. The proof can be found in Section E.3.

Using the estimation error in Theorem 5 and approximation error in Theorem 9, by choosing $\bar{M}$ to minimax the total error, we can conclude the sample complexity of parallel neural networks using weight decay, which is the main result (Theorem 1) of this paper. See Section F for the detail.

# 5 Experiment

We empirically compare a parallel neural network (PNN) and a vanilla ReLU neural network (NN) with smoothing spline, trend filtering (TF) [40], and wavelet denoising. Trend filtering can be viewed as a more efficient discrete spline version of locally adaptive regression spline and enjoys the same optimal rates for the BV classes. Wavelet denoising is also known to be minimax-optimal for the BV classes. The results are shown in Figure 2. We use two target functions: a Doppler function whose frequency is decreasing(Figure 2(a)-(c)), and a combination of piecewise linear function and piecewise cubic function, or "vary" function (Figure 2(d)-(f)). We repeat each experiment 10 times and take the average. The shallow area in Figure 2(b)(e) shows 95% confidence interval by inverting the Wald's test. The degree of freedom (DoF) is computed based on Tibshirani [41].

As can be shown in the figure, both TF and wavelet denoising can adapt to the different levels of smoothness in the target function, while smoothing splines tend to be oversmoothed where the target function is less smooth (the left side in (a)(d), enlarged in (g)). The prediction of PNN is similar to TF and wavelet denoising and shows local adaptivity. Besides, the MSE of PNN almost follows the same trend as TF and wavelet denoising which is consistent with our theoretical understanding that the error rate of neural network is closer to locally adaptive methods. Notably PNN, TF and wavelet denoising achieve lower error at a much smaller degree-of-freedom than smoothing splines.

In a vanilla NN, weight decay is equivalent to $\ell_1$ regularizer in any two successive layers, but to the best of our knowledge it does not lead to sparse representation learning unless some specific sparse structure is enforced. While our theory does not apply to vanilla neural networks, the results seem to suggest the NN behaves similar to smoothing spline and is *not* locally adaptive.

There are some mild drops in the best MSE one can achieve with NN vs TF in both examples. We are surprised that the drop is small because NN needs to learn the basis functions that TF essentially hard-coded. The additional price to pay for using a more adaptive and more flexible representation learning method seems not high at all.

In Figure 2(c)(f), we give the output all the "active" subnetwork, i.e. the subnetworks whose output is not a constant. Notice that the number of active subnetworks is much smaller than the initialization. This is because weight decay induces $\ell_p$ sparsity and the weight in most of the subnetworks reduces towards 0 after training. More details are shown in Section G.

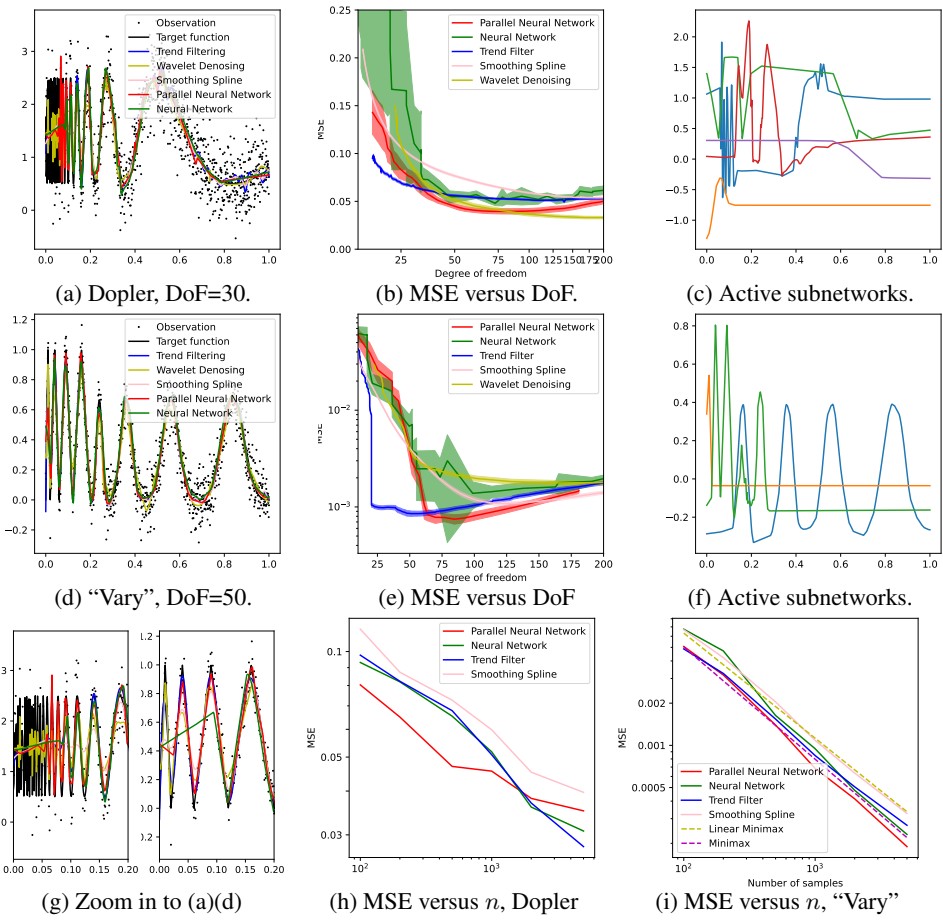

Figure 2: Numerical experiment results of the Doppler function (a-c,h), and "vary" function (d-f,g). All the "active" subnetworks are plotted in (c)(f). The horizontal axis in (b) is not linear.

In Figure 2(h)(i), we plot the MSE versus the number of training samples for "Doppler" and "Vary" respectively. It is clear that parallel NN works the best overall. In (i), we further compare the scaling of the MSE against the minimax rate ($n^{-4/5}$) and the minimax linear rate ($n^{-3/4}$), i.e., the best rate kernel methods could achieve. As is predicted by our theory, when $n$ is large, the MSE of parallel neural networks and trend filtering decreases at almost the same rate as the minimax rate, while smoothing splines, as expected, is converging at the (suboptimal) minimax linear rate. Interestingly, vanilla NN seems to converge at the optimal rate too on this example. It remains an open question whether vanila NN is merely "lucky" on this example, or it also achieves the minimax rate for all functions in BV(m).

## 6  Conclusion and Discussion

In this paper, we show that a deep parallel neural network can be locally adaptive by tuning only the weight decay parameter. This confirms that neural networks can be nearly optimal in learning functions with heterogeneous smoothness which separates them from kernel methods. We prove that training an $L$ layer parallel neural network with weight decay is equivalent to an $\ell_{2/L}$-penalized regression model with representation learning. Since in typical application $L \gg 2$, weight decay promotes a sparse linear combination of the learned bases. Using this method, we proved that a parallel neural network can achieve close to the minimax rate in the Besov space and bounded variation (BV) space. Our result reveals that one do not need to specify the smoothness parameter $\alpha$ (or $m$). Neural networks can adapt to different degree of smoothness, or choose different parameters for different regions of the domain of the target function. This is a new type of adaptivity not

possessed by traditional adaptive nonparametric regression methods like locally adaptive regression spline or trend filtering.

On the other hand, as the depth of neural network $L$ increases, $2/L$ tends to 0 and the error rate moves closer to the minimax rate of Besov and BV space. This indicates that when the sample size is large enough, deeper models have smaller error than shallower models, and helps explain why empirically deep neural networks has better performance than shallow neural networks.

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

## A  Other related works

Besides Parhi and Nowak [29] which we discussed earlier, Parhi and Nowak [30, 31] also leveraged the connections between NNs and splines. Parhi and Nowak [30] focused on characterizing the variational form of multi-layer NN. Parhi and Nowak [31] showed that two-layer ReLU activated NN achieves minimax rate for a BV class of order 1 but did not cover multilayer NNs nor BV class with order $> 1$, which is our focus.

The connection between weight-decay regularization with sparsity-inducing penalties in two-layer NNs is folklore and used by Neyshabur et al. [27], Savarese et al. [35], Ongie et al. [28], Ergen and Pilanci [13, 16], Parhi and Nowak [29, 31], Pilanci and Ergen [32]. The key underlying technique — an application of the AM-GM inequality (which we used in this paper as well) — can be traced back to Srebro et al. [38] (see a recent exposition by Tibshirani [42]). [42] also generalized the result to multi-layered NNs, but with a simple (element-wise) connections. [14] generalized the results to a three-layer parallel neural network, and proved its equivalence to an $\ell_1$ sparse model, but this requires a non-standard regularizer. Besides, [12] proved that training a two-layer convolution neural network (CNN) with weight decay induces sparsity, and points to a potential extension to these works including our work.

The approximation-theoretic and estimation-theoretic research for neural network has a long history too [6, 4, 49, 36, 39]. Most existing work considered the Holder, Sobolev spaces and their extensions, which contain only homogeneously smooth functions and cannot demonstrate the advantage of NNs over kernels. The only exception is Suzuki [39] which, as we discussed earlier, requires modifications to NN architecture for each class. In contrast, we require tuning only the standard weight decay parameter.

## B  Two-layer Neural Network with Truncated Power Activation Functions

We start by recapping the result of Parhi and Nowak [29] and formalizing its implication in estimating BV functions. Parhi and Nowak [29] considered a two layer neural network with truncated power activation function. Let the neural network be

$$f(x) = \sum_{j=1}^{M} v_j \sigma^m(w_j x + b_j) + c(x), \tag{7}$$

where $w_j, v_j$ denote the weight in the first and second layer respectively, $b_j$ denote the bias in the first layer, $c(x)$ is a polynomial of order up to $m$, $\sigma^m(x) := \max(x, 0)^m$. Parhi and Nowak [29, Theorem 8] showed that when $M$ is large enough, The optimization problem

$$\min_{\boldsymbol{w}, \boldsymbol{v}} \hat{L}(f) + \frac{\lambda}{2} \sum_{j=1}^{M} (|v_j|^2 + |w_j|^{2m}) \tag{8}$$

is equivalent to the locally adaptive regression spline:

$$\min_{f} \hat{L}(f) + \lambda TV(f^{(m)}(x)), \tag{9}$$

which optimizes over arbitrary functions that is $m$-times weakly differentiable. The latter was studied in Mammen and van de Geer [24], which leads to the following MSE:

**Theorem 10.** *Let $M \geq n - m$, and $\hat{f}$ be the function (7) parameterized by the minimizer of* (8)*, then*

$$\mathrm{MSE}(\hat{f}) = O(n^{-(2m+2)(2m+3)}).$$

We show a simpler proof in the univariate case due to Tibshirani [43]:

*Proof.* As is shown in Parhi and Nowak [29, Theorem 8], the minimizer of (8) satisfy

$$|v_j| = |w_j|^m, \forall k$$

so the TV of the neural network $f_{NN}$ is

$$TV^{(m)}(f_{NN}) = TV^{(m)}c(x) + \sum_{j=1}^{M}|v_j||w_j|^m TV^{(m)}(\sigma^{(m)}(x))$$

$$= \sum_{j=1}^{M}|v_j||w_j|^m$$

$$= \frac{1}{2}\sum_{j=1}^{M}(|v_j|^2 + |w_j|^{2m})$$

which shown that (8) is equivalent to the locally adaptive regression spline (9) as long as the number of knots in (9) is no more than $M$. Furthermore, it is easy to check that any spline with knots no more than $M$ can be expressed as a two layer neural network (8). It suffices to prove that the solution in (9) has no more than $n - m$ number of knots.

Mammen and van de Geer [24, Proposition 1] showed that there is a solution to (9) $\hat{f}(x)$ such that $\hat{f}(x)$ is a $m$th order spline with a finite number of knots but did not give a bound. Let the number of knots be $M$, we can represent $\hat{f}$ using the truncated power basis

$$\hat{f}(x) = \sum_{j=1}^{M}a_j(x - t_j)^m_+ + c(x) := \sum_{j=1}^{M}a_j\sigma_j^{(m)}(x) + c(x)$$

where $t_j$ are the knots, $c(x)$ is a polynomial of order up to $m$, and define $\sigma_j^{(m)}(x) = (x - t_j)^m_+$.

Mammen and van de Geer [24] however did not give a bound on $M$. Parhi and Nowak [29]'s Theorem 1 implies that $M \leq n - m$. Its proof is quite technical and applies more generally to a higher dimensional generalization of the BV class.

Tibshirani [43] communicated to us the following elegant argument to prove the same using elementary convex analysis and linear algebra, which we present below.

Define $\Pi_m(f)$ as the $L^2(P_n)$ projection of $f$ onto polynomials of degree up to $m$, $\Pi_m^\perp(f) := f - \Pi_m(f)$. It is easy to see that

$$\Pi_m^\perp f(x) = \sum_{j=1}^{M}a_j\Pi_m^\perp\sigma_j^{(m)}(x)$$

Denote $f(x_{1:n}) := \{f(x_1), \ldots, f(x_n)\} \in \mathbb{R}^n$ as a vector of all the predictions at the sample points.

$$\Pi_m^\perp \hat{f}(x_{1:n}) = \sum_{j=1}^{M}a_j\Pi_m^\perp\sigma_j^{(m)}(x_{1:n}) \in \Pi_m^\perp \text{conv}\{\pm\sigma_j^{(m)}(x_{1:n})\}\cdot\sum_{j=1}^{M}|a_j| =\in \text{conv}\{\pm\Pi_m^\perp\sigma_j^{(m)}(x_{1:n})\}\cdot\sum_{j=1}^{M}|a_j|$$

where conv denotes the convex hull of a set. The convex hull $\text{conv}\{\pm\sigma_j^{(m)}(x_{1:n})\}\cdot\sum_{j=1}^{M}|a_j|$ is an $n$-dimensional space, and polynomials of order up to $m$ is an $m + 1$ dimensional space, so the set defined above has dimension $n - m - 1$. By Carathéodory's theorem, there is a subset of points in this space

$$\{\Pi_m^\perp\sigma_{j_k}^{(m)}(x_{1:n})\} \subseteq \{\Pi_m^\perp\sigma_j^{(m)}(x_{1:n})\}, 1 \leq k \leq n - m$$

such that

$$\Pi_m^\perp f(x) = \sum_{k=1}^{n-m}\tilde{a}_k\Pi_m^\perp\sigma_{j_k}^{(m)}(x), \sum_{k=1}^{n-m}|a_k| \leq 1$$

In other word, there exist a subset of knots $\{\tilde{t}_j, j \in [n - m]\}$ that perfectly recovers $\Pi_m^\perp\hat{f}(x)$ at all the sample points, and the TV of this function is no larger than $\hat{f}$.

This shows that

$$\tilde{f}(x) = \sum_{j=1}^{n-m}\tilde{a}_j(x - t_j)^m_+, s.t.\tilde{f}(x_i) = f(x_i)$$

552 for all $x_i$ in $n$ onbservation points.

553 The MSE of locally adaptivity regressive spline (9) was studied in Mammen and van de Geer [24,
554 Section 3], which equals the error rate given in Theorem 10. □

555 This indicates that the neural network (7) is minimax optimal for $BV(m)$.

Let us explain a few the key observations behind this equivalence. (a) The truncated power functions
(together with an $m$th order polynomial) spans the space of an $m$th order spline. (b) The neural
network in (7) is equivalent to a free-knot spline with $M$ knots (up to reparameterization). (c) A
solution to (9) is a spline with at most $n - m$ knots [29, Theorem 8]. (d) Finally, by the AM-GM
inequality

$$|v_j|^2 + |w_j|^{2m} \geq 2|v_j||w_j|^m = 2|c_j|$$

556 where $c_j = v_j|w_j|^m$ is the coefficient of the corresponding $j$th truncated power basis. The $m$th
557 order total variation of a spline is equal to $\sum_j |c_j|$. It is not hard to check that the loss function
558 depends only on $c_j$, thus the optimal solution will always take "=" in the AM-GM inequality.

## C  Introduction To Common Function Classes

560 In the following definition define $\Omega$ be the domain of the function classes, which will be omitted in
561 the definition.

### C.1  Besov Class

563 **Definition 1.** *Modulus of smoothness: For a function $f \in L^p(\Omega)$ for some $1 \leq p \leq \infty$, the $r$-th
564 modulus of smoothness is defined by*

$$w_{r,p}(f,t) = \sup_{h \in \mathbb{R}^d : \|h\|_2 \leq t} \|\Delta_h^r(f)\|_p,$$

565

$$\Delta_h^r(f) := \begin{cases} \sum_{j=0}^{r} \binom{r}{j}(-1)^{r-j} f(x+jh), & \text{if } x \in \Omega, x + rh \in \Omega, \\ 0, & \text{otherwise.} \end{cases}$$

566 **Definition 2.** *Besov space: For $1 \leq p, q \leq \infty, \alpha > 0, r := \lceil \alpha \rceil + 1$, define*

$$|f|_{B_{p,q}^\alpha} = \begin{cases} \left( \int_{t=0}^{\infty} (t^{-\alpha} w_{r,p}(f,t))^q \frac{dt}{t} \right)^{\frac{1}{q}}, & q < \infty \\ \sup_{t>0} t^{-\alpha} w_{r,p}(f,t), & q = \infty, \end{cases}$$

567 *and define the norm of Besov space as:*

$$\|f\|_{B_{p,q}^\alpha} = \|f\|_p + |f|_{B_{p,q}^\alpha}.$$

568 *A function $f$ is in the Besov space $B_{p,q}^\alpha$ if $\|f\|_{B_{p,q}^\alpha}$ is finite.*

569 Note that the Besov space for $0 < p, q < 1$ is also defined, but in this case it is a quasi-Banach space
570 instead of a Banach space and will not be covered in this paper.

571 Functions in Besov space can be decomposed using B-spline basis functions. Any function $f$ in
572 Besov space $B_{p,q}^\alpha, \alpha > d/p$ can be decomposed using B-spline of order $m, m > \alpha$: let $\boldsymbol{x} \in \mathbb{R}^d$,

$$f(\boldsymbol{x}) = \sum_{k=0}^{\infty} \sum_{\boldsymbol{s} \in J(k)} c_{k,\boldsymbol{s}}(f) M_{m,k,\boldsymbol{s}}(\boldsymbol{x}) \tag{10}$$

573 where $J(k) := \{2^{-k}\boldsymbol{s} : \boldsymbol{s} \in [-m, 2^k + m]^d \subset \mathbb{Z}^d\}$, $M_{m,k,\boldsymbol{s}}(\boldsymbol{x}) := M_m(2^k(\boldsymbol{x} - \boldsymbol{s}))$, and $M_k(\boldsymbol{x}) =$
574 $\prod_{i=1}^{d} M_k(x_i)$ is the cardinal B-spline basis function which can be expressed as a polynomial:

$$M_m(x) = \frac{1}{m!} \sum_{j=1}^{m+1} (-1)^j \binom{m+1}{j} (x-j)_+^m$$

$$= ((m+1)/2)^m \frac{1}{m!} \sum_{j=1}^{m+1} (-1)^j \binom{m+1}{j} \left( \frac{x-j}{(m+1)/2} \right)_+^m, \tag{11}$$

Furthermore, the norm of Besov space is equivalent to the sequence norm:

$$\|\{c_{k,\boldsymbol{s}}\}\|_{b_{p,q}^\alpha} := \Big(\sum_{k=0}^\infty (2^{(\alpha-d/p)k}\|\{c_{k,\boldsymbol{s}}(f)\}_{\boldsymbol{s}}\|_p)^q\Big)^{1/q} \eqsim \|f\|_{B_{p,q}^\alpha}.$$

See e.g. Dũng [11, Theorem 2.2] for the proof.

The Besov space is closely connected to other function spaces including the Hölder space ($\mathcal{C}^\alpha$) and the Sobolev space ($W_p^\alpha$). Specifically, if the domain of the functions is $d$-dimensional [39, 34],

- $\forall \alpha \in \mathbb{N}, B_{p,1}^\alpha \subset W_p^\alpha \subset B_{p,\infty}^\alpha$, and $B_{2,2}^\alpha = W_2^\alpha$.
- For $0 < \alpha < \infty$ and $\alpha \in \mathcal{N}, \mathcal{C}^\alpha = B_{\infty,\infty}^\alpha$.
- If $\alpha > d/p$, $B_{p,q}^\alpha \subset \mathcal{C}^0$.

## C.2 Other Function Spaces

**Definition 3.** *Hölder space: let $m \in \mathbb{N}$, the $m$-th order Holder class is defined as*

$$\mathcal{C}^m = \left\{ f : \max_{|a|=k} \frac{|D^a f(x) - D^a f(z)|}{\|x-z\|_2} < \infty, \forall x, z \in \Omega \right\}$$

*where $D^a$ denotes the weak derivative.*

Note that fraction order of Hölder space can also be defined. For simplicity, we will not cover that case in this paper.

**Definition 4.** *Sobolev space: let $m \in \mathcal{N}, 1 \le p \le \infty$, the Sobolev norm is defined as*

$$\|f\|_{W_p^m} := \left( \sum_{|a| \le m} \|D^a f\|_p^p \right)^{1/p},$$

*the Sobolev space is the set of functions with finite Sobolev norm:*

$$W_p^m := \{f : \|f\|_{W_p^m} < \infty\}.$$

**Definition 5.** *Total Variation (TV): The total variation (TV) of a function $f$ on an interval $[a,b]$ is defined as*

$$TV(f) = \sup_{\mathcal{P}} \sum_{i=1}^{n_\mathcal{P}-1} |f(x_{i+1}) - f(x_i)|$$

*where the $\mathcal{P}$ is taken among all the partitions of the interval $[a,b]$.*

In many applications, functions with stronger smoothness conditions are needed, which can be measured by high order total variation.

**Definition 6.** *High order total variation: the $m$-th order total variation is the total variation of the $(m-1)$-th order derivative*

$$TV^{(m)}(f) = TV(f^{(m-1)})$$

**Definition 7.** *Bounded variation (BV): The $m$-th order bounded variation class is the set of functions whose total variation (TV) is bounded.*

$$BV(m) := \{f : TV(f^{(m)}) < \infty\}.$$

# D  Proof of Estimation Error

## D.1 Equivalence Between Parallel Neural Networks and $p$-norm Penalized Problems

**Proposition 4.** *Fix the input dataset $\mathcal{D}_n$ and a constant $c_1 > 0$. There exists an one-to-one mapping between $\lambda > 0$ and $P' > 0$ such that (2) is equivalent to the following problem:*

$$\underset{\{\bar{\mathbf{W}}_j^{(\ell)}, \bar{\boldsymbol{b}}_j^{(\ell)}, a_j\}}{\arg\min} \hat{L}\Big(\sum_{j=1}^M a_j \bar{f}_j\Big) = \frac{1}{n} \sum_i (y_i - \bar{f}_{1:M}(\boldsymbol{x}_i)^T \boldsymbol{a})^2$$

$$s.t.\ \|\bar{\mathbf{W}}_j^{(1)}\|_F \le c_1\sqrt{d}, \forall j \in [M],$$

$$\|\bar{\mathbf{W}}_j^{(\ell)}\|_F \le c_1\sqrt{w}, \forall j \in [M], 2 \le \ell \le L, \qquad \|\{a_j\}\|_{2/L}^{2/L} \le P'$$

602   *where $\bar{f}_j(\cdot)$ is a subnetwork with parameters $\bar{\mathbf{W}}_j^{(\ell)}, \bar{\boldsymbol{b}}_j^{(\ell)}$.*

603   *Proof.* Using Lagrange's method, one can easily find (2) is equivalent to a constrained optimization
604   problem:

$$\underset{\{\mathbf{W}_j^{(\ell)}, \boldsymbol{b}_j^{(\ell)}\}}{\arg\min} \hat{L}\Big(\sum_{j=1}^M f_j\Big), \quad s.t. \sum_{j=1}^M \sum_{\ell=1}^L \|\mathbf{W}_j^{(\ell)}\|_F^2 \leq P \tag{12}$$

605   for some constant $P$ that depends on $\lambda$ and the dataset $\mathcal{D}$.

606   We make use of the property from (4) to minimize the constraint term in (12) while keeping this
607   neural network equivalent to the original one. Specifically, let $\mathbf{W}^{(1)}, \boldsymbol{b}^{(1)}, \dots \mathbf{W}^{(L)}, \boldsymbol{b}^{(L)}$ be the
608   parameters of an $L$-layer neural network.

$$f(x) = \mathbf{W}^{(L)} \sigma(\mathbf{W}^{(L-1)} \sigma(\dots \sigma(\mathbf{W}^{(1)} x + \boldsymbol{b}^{(1)}) \dots) + \boldsymbol{b}^{(L-1)}) + \boldsymbol{b}^{(L)},$$

609   which is equivalent to

$$f(x) = \alpha_L \tilde{\mathbf{W}}^{(L)} \sigma(\alpha_{L-1} \tilde{\mathbf{W}}^{(L-1)} \sigma(\dots \sigma(\alpha_1 \tilde{\mathbf{W}}^{(1)} x + \tilde{\boldsymbol{b}}^{(1)}) \dots) + \tilde{\boldsymbol{b}}^{(L-1)}) + \tilde{\boldsymbol{b}}^{(L)},$$

610   as long as $\alpha_\ell > 0, \prod_{\ell=1}^L \alpha^L = \prod_{\ell=1}^L \|\mathbf{W}^{(\ell)}\|_F$, where $\tilde{\mathbf{W}}^{(\ell)} := \frac{\mathbf{W}^{(\ell)}}{\|\mathbf{W}^{(\ell)}\|_F}$. By the AM-GM inequal-
611   ity, the $\ell_2$ regularizer of the latter neural network is

$$\sum_{\ell=1}^L \|\alpha_\ell \tilde{\mathbf{W}}^{(\ell)}\|_F^2 = \sum_{\ell=1}^L \alpha_\ell^2 \geq L \left(\prod_{\ell=1}^L a_\ell\right)^{2/L} = L \left(\prod_{\ell=1}^L \|\mathbf{W}^{(\ell)}\|_F\right)^{2/L}$$

612   and equality is reached when $\alpha_1 = \alpha_2 = \dots = \alpha_L$. In other word, in the problem (2), it suffices to
613   consider the network that satisfies

$$\|\mathbf{W}_j^{(1)}\|_F = \|\mathbf{W}_j^{(2)}\|_F = \dots = \|\mathbf{W}_j^{(L)}\|_F, \forall j \in [M], \ell \in [L]. \tag{13}$$

614   Using (4) again, one can find that the neural network is also equivalent to

$$f(x) = \sum_{j=1}^M a_j \bar{\mathbf{W}}^{(L)} \sigma(\bar{\mathbf{W}}_j^{(L-1)} \sigma(\dots \sigma(\bar{\mathbf{W}}_j^{(1)} x + \bar{\boldsymbol{b}}_j^{(1)}) \dots) + \bar{\boldsymbol{b}}_j^{(L-1)}) + \bar{\boldsymbol{b}}_j^{(L)},$$

615   where

$$\|\bar{\mathbf{W}}_j^{(\ell)}\|_F \leq \beta^{(\ell)}, a_j = \frac{\prod_{\ell=1}^L \|\mathbf{W}_j^{(\ell)}\|_F}{\prod_{\ell=1}^L \beta^{(\ell)}} = \frac{\|\mathbf{W}_j^{(1)}\|_F^L}{\prod_{\ell=1}^L \beta^{(\ell)}} = \frac{(\sum_{\ell=1}^L \|\mathbf{W}_j^{(\ell)}\|_F^2 / L)^{L/2}}{\prod_{\ell=1}^L \beta^{(\ell)}}, \tag{14}$$

616   where the last two equality comes from the assumption (13). Choosing $\beta^{(\ell)} = c_1 \sqrt{w}$ expect $\ell = 1$
617   where $\beta^{(1)} = c_1 \sqrt{d}$, and scaling $\bar{\boldsymbol{b}}^{(\ell)}$ accordingly and taking the constraint in (12) into (14) finishes
618   the proof. $\qquad\square$

## D.2   Covering Number of Parallel Neural Networks

620   **Theorem 5.** *The covering number of the model defined in (5) apart from the bias in the last layer*
621   *satisfies*

$$\log \mathcal{N}(\mathcal{F}, \delta) \lesssim w^{2+2/(1-2/L)} L^2 \sqrt{d} P'^{\frac{1}{1-2/L}} \delta^{-\frac{2/L}{1-2/L}} \log(wP'/\delta).$$

622

The proof relies on the covering number of each subnetwork in a parallel neural network
(Lemma 11), observing that $|f(x)| \leq 2^{L-1} w^{L-1} \sqrt{d}$ under the condition in Lemma 11, and
then apply Lemma 6. We argue that our choice of condition on $\|\boldsymbol{b}^{(\ell)}\|_2$ in Lemma 11 is suf-
ficient to analyzing the model apart from the bias in the last layer, because it guarantees that
$\sqrt{w}\|\mathbf{W}^{(\ell)} \mathcal{A}_{\ell-1}(x)\|_2 \leq \|\boldsymbol{b}^{(\ell)}\|_2$. This leads to

$$\|\mathbf{W}^{(\ell)} \mathcal{A}_{\ell-1}(\boldsymbol{x})\|_\infty \leq \|\mathbf{W}^{(\ell)} \mathcal{A}_{\ell-1}(\boldsymbol{x})\|_2 \leq \sqrt{w}\|\boldsymbol{b}^{(\ell)}\|_2 \leq \|\boldsymbol{b}^{(\ell)}\|_\infty$$

623 If this condition is not met, $\mathbf{W}^{(\ell)}\mathcal{A}_{\ell-1}(\boldsymbol{x}) + b^{(\ell)}$ is either always positive or always negative
624 for all feasible $\boldsymbol{x}$ along at least one dimension. If $(\mathbf{W}^{(\ell)}\mathcal{A}_{\ell-1}(\boldsymbol{x}) + b^{(\ell)})_i$ is always negative,
625 one can replace $b^{(\ell)})_i$ with $-\max_{\boldsymbol{x}}\|\mathbf{W}^{(\ell)}\mathcal{A}_{\ell-1}(\boldsymbol{x})\|_{\infty}$ without changing the output of this model
626 for any feasible $\boldsymbol{x}$. If $(\mathbf{W}^{(\ell)}\mathcal{A}_{\ell-1}(\boldsymbol{x}) + b^{(\ell)})_i$ is always positive, one can replace $b^{(\ell)})_i$ with
627 $\max_{\boldsymbol{x}}\|\mathbf{W}^{(\ell)}\mathcal{A}_{\ell-1}(\boldsymbol{x})\|_{\infty}$, and adjust the bias in the next layer such that the output of this model
628 is not changed for any feasible $\boldsymbol{x}$. In either cases, one can replace the bias $\boldsymbol{b}^{(\ell)}$ with another one with
629 smaller norm while keeping the model equivalent except the bias in the last layer.

630 **Lemma 11.** *Let $\mathcal{F} \subseteq \{f : R^d \to \mathbb{R}\}$ denote the set of L-layer neural network (or a subnetwork in*
631 *a parallel neural network) with width $w$ in each hidden layer. It has the form*

$$f(x) = \mathbf{W}^{(L)}\sigma(\mathbf{W}^{(L-1)}\sigma(\dots\sigma(\mathbf{W}^{(1)}x + \boldsymbol{b}^{(1)})\dots) + \boldsymbol{b}^{(L-1)}) + \boldsymbol{b}^{(L)},$$
$$\mathbf{W}^{(1)} \in \mathbb{R}^{w \times d}, \|\mathbf{W}^{(1)}\|_F \le \sqrt{d}, \boldsymbol{b}^{(1)} \in \mathbb{R}^w, \|\boldsymbol{b}^{(1)}\|_2 \le \sqrt{dw},$$
$$\mathbf{W}^{(\ell)} \in \mathbb{R}^{w \times w}\|\mathbf{W}^{(\ell)}\|_F \le \sqrt{w}, \boldsymbol{b}^{(\ell)} \in \mathbb{R}^w, \|\boldsymbol{b}^{(\ell)}\|_2 \le 2^{\ell-1}w^{\ell-1}\sqrt{dw}, \quad \forall \ell = 2, \dots L-1,$$
$$\mathbf{W}^{(L)} \in \mathbb{R}^{1 \times w}, \|\mathbf{W}^{(L)}\|_F \le \sqrt{w}, b^{(L)} = 0$$
$$(15)$$

632 *and $\sigma(\cdot)$ is the ReLU activation function, the input satisfy $\|x\|_2 \le 1$, then the supremum norm*
633 *$\delta$-covering number of $\mathcal{F}$ obeys*

$$\log \mathcal{N}(\mathcal{F}, \delta) \le c_7 L w^2 \log(1/\delta) + c_8$$

634 *where $c_7$ is a constant depending only on $d$, and $c_8$ is a constant that depend on $d, w$ and $L$.*

635 *Proof.* First study two neural networks which differ by only one layer. Let $g_\ell, g'_\ell$ be two neural net-
636 works satisfying (15) with parameters $\mathbf{W}_1, \boldsymbol{b}_1, \dots, \mathbf{W}_L, \boldsymbol{b}_L$ and $\mathbf{W}'_1, \boldsymbol{b}'_1, \dots, \mathbf{W}'_L, \boldsymbol{b}'_L$ respectively.
637 Furthermore, the parameters in these two models are the same except the $\ell$-th layer, which satisfy

$$\|\mathbf{W}_\ell - \mathbf{W}'_\ell\|_F \le \epsilon, \|\boldsymbol{b}_\ell - \boldsymbol{b}'_\ell\|_2 \le \tilde{\epsilon}.$$

638 Denote the model as

$$g_\ell(x) = \mathcal{B}_\ell(\mathbf{W}_\ell \mathcal{A}_\ell(\boldsymbol{x}) + \boldsymbol{b}_\ell), g'_\ell(x) = \mathcal{B}_\ell(\mathbf{W}'_\ell \mathcal{A}_\ell(\boldsymbol{x}) + \boldsymbol{b}'_\ell)$$

639 where $\mathcal{A}_\ell(\boldsymbol{x}) = \sigma(\mathbf{W}_{\ell-1}\sigma(\dots\sigma(\mathbf{W}_1 x + \boldsymbol{b}_1)\dots) + \boldsymbol{b}_{\ell-1})$ denotes the first $\ell-1$ layers in the neural
640 network, and $\mathcal{A}_\ell(x) = \mathbf{W}_L\sigma(\dots\sigma(\mathbf{W}_{\ell+1}\sigma(x) + \boldsymbol{b}_{\ell+1})\dots) + \boldsymbol{b}_L)$ denotes the last $L - \ell - 1$ layers,
641 with definition $\mathcal{A}_1(\boldsymbol{x}) = \boldsymbol{x}, \mathcal{B}_L(\boldsymbol{x}) = \boldsymbol{x}$.

642 Now focus on bounding $\|\mathcal{A}(\boldsymbol{x})\|$. Let $\mathbf{W} \in \mathbb{R}^{m \times m'}, \|\mathbf{W}\|_F \le \sqrt{m'}, \boldsymbol{x} \in \mathbb{R}^{m'}, \boldsymbol{b} \in \mathbb{R}^m, \|\boldsymbol{b}\|_2 \le$
643 $\sqrt{m}$

$$\begin{aligned}\|\sigma(\mathbf{W}\boldsymbol{x} + \boldsymbol{b})\|_2 &\le \|\mathbf{W}\boldsymbol{x} + \boldsymbol{b}\|_2 \\ &\le \|\mathbf{W}\|_2\|\boldsymbol{x}\|_2 + \|\boldsymbol{b}\|_2 \\ &\le \|\mathbf{W}\|_F\|\boldsymbol{x}\|_2 + \|\boldsymbol{b}\|_2 \\ &\le \sqrt{m'}\|\boldsymbol{x}\|_2 + \sqrt{m}\end{aligned}$$

644 where we make use of $\|\cdot\|_2 \le \|\cdot\|_F$. Because of that,

$$\begin{aligned}\|\mathcal{A}_2(\boldsymbol{x})\|_2 &\le \sqrt{d} + \sqrt{dw} \le 2\sqrt{dw}, \\ \|\mathcal{A}_3(\boldsymbol{x})\|_2 &\le \sqrt{w}\|\mathcal{A}_2(\boldsymbol{x})\|_2 + 2w\sqrt{dw} \le 4w\sqrt{dw}, \\ &\dots \\ \|\mathcal{A}_\ell(\boldsymbol{x})\|_2 &\le \sqrt{w}\|\mathcal{A}_{\ell-1}(\boldsymbol{x})\|_2 \le 2\sqrt{dw}(2w)^{\ell-2}.\end{aligned} \quad (16)$$

645 Then focus on $\mathcal{B}(\boldsymbol{x})$. Let $\mathbf{W} \in \mathbb{R}^{m \times m'}, \|\mathbf{W}\|_F \le \sqrt{m'}, \boldsymbol{x}, \boldsymbol{x}' \in \mathbb{R}^{m'}, \boldsymbol{b} \in \mathbb{R}^m, \|\boldsymbol{b}\|_2 \le \sqrt{m}$.
646 Furthermore, $\|\boldsymbol{x} - \boldsymbol{x}'\|_2 \le \epsilon$, then

$$\|\sigma(\mathbf{W}\boldsymbol{x} + \boldsymbol{b}) - \sigma(\mathbf{W}\boldsymbol{x}' + \boldsymbol{b})\|_2 \le \|\mathbf{W}(\boldsymbol{x} - \boldsymbol{x}')\|_2 \le \|\mathbf{W}\|_F\|\boldsymbol{x} - \boldsymbol{x}'\|_2$$

647 which indicates that $\|\mathcal{B}(\boldsymbol{x}) - \mathcal{B}(\boldsymbol{x})'\|_2 \le (\sqrt{w})^{L-\ell}\|\boldsymbol{x} - \boldsymbol{x}'\|_2$

648 Finally, for any $\mathbf{W}, \mathbf{W}' \in \mathbb{R}^{m \times m'}, \boldsymbol{x} \in \mathbb{R}^{m'}, \boldsymbol{b}, \boldsymbol{b}' \in \mathbb{R}^m$, one have

$$\|(\mathbf{W}\boldsymbol{x} + \boldsymbol{b}) - (\mathbf{W}'\boldsymbol{x} + \boldsymbol{b}')\|_2 = \|(\mathbf{W} - \mathbf{W}')\boldsymbol{x} + (\boldsymbol{b} - \boldsymbol{b}')\|_2$$
$$\leq \|\mathbf{W} - \mathbf{W}'\|_2 \|\boldsymbol{x}\|_2 + \|\boldsymbol{b} - \boldsymbol{b}'\|_2.$$
$$\leq \|\mathbf{W} - \mathbf{W}'\|_F \|\boldsymbol{x}\|_2 + \sqrt{m}\|\boldsymbol{b} - \boldsymbol{b}'\|_\infty.$$

649 In summary,

$$|g_\ell(\boldsymbol{x}) - g'_\ell(\boldsymbol{x})| = |\mathcal{B}_\ell(\mathbf{W}_\ell \mathcal{A}_\ell(\boldsymbol{x}) + \boldsymbol{b}_\ell) - \mathcal{B}_\ell(\mathbf{W}'_\ell \mathcal{A}_\ell(\boldsymbol{x}) + \boldsymbol{b}'_\ell)|$$
$$\leq (\sqrt{w})^{L-\ell} \|(\mathbf{W}_\ell \mathcal{A}_\ell(\boldsymbol{x}) + \boldsymbol{b}_\ell) - (\mathbf{W}'_\ell \mathcal{A}_\ell(\boldsymbol{x}) + \boldsymbol{b}'_\ell)\|_2$$
$$\leq (\sqrt{w})^{L-\ell} (\|\mathbf{W}_\ell - \mathbf{W}'_\ell\|_F \|\mathcal{A}_\ell(\boldsymbol{x})\|_2 + \|\boldsymbol{b}_\ell - \boldsymbol{b}'_\ell\|_2)$$
$$\leq 2^{(\ell-1)} w^{(L+\ell-3)/2} d^{1/2} \epsilon + w^{(L-\ell)/2} \bar{\epsilon}$$

650 Let $f(x), f'(x)$ be two neural networks satisfying (15) with parameters $W_1, b_1, \ldots, W_L, b_L$ and
651 $W'_1, b'_1, \ldots, W'_L, b'_L$ respectively, and $\|W_\ell - W'_\ell\|_F \leq \epsilon_\ell, \|b_\ell - b'_\ell\|_F \leq \tilde{\epsilon}_\ell$. Further define $f_\ell$ be the
652 neural network with parameters $W_1, b_1, \ldots, W_\ell, b_\ell, W'_{\ell+1}, b'_{\ell+1}, \ldots, W'_L, b'_L$, then

$$|f(x) - f'(x)| \leq |f(x) - f_1(x)| + |f_1(x) - f_2(x)| + \cdots + |f_{L-1}(x) - f'(x)|$$
$$\leq \sum_{\ell=1}^{L} 2^{(\ell-2)} d^{1/2} w^{(L+\ell-3)/2} \epsilon + w^{(L-\ell)/2} \bar{\epsilon}$$

For any $\delta > 0$, one can choose

$$\epsilon_\ell = \frac{\delta}{2^\ell w^{(L+\ell-3)/2} d^{1/2}}, \tilde{\epsilon}_\ell = \frac{\delta}{2 w^{(L-\ell)/2}}$$

653 such that $|f(x) - f'(x)| \leq \delta$.

654 On the other hand, the $\epsilon$-covering number of $\{\mathbf{W} \in \mathbb{R}^{m \times m'} : \|\mathbf{W}\|_F \leq \sqrt{m'}\}$ on Frobenius norm
655 is no larger than $(2\sqrt{m'}/\epsilon + 1)^{m \times m'}$, and the $\bar{\epsilon}$-covering number of $\{\boldsymbol{b} \in \mathbb{R}^m : \|\boldsymbol{b}\|_2 \leq 1\}$ on
656 infinity norm is no larger than $(2/\bar{\epsilon} + 1)^m$. The entropy of this neural network can be bounded by

$$\log \mathcal{N}(f; \delta) \leq w^2 L \log(2^{L+1} w^{L-1}/\delta + 1) + wL \log(2^{L-1} w^{(L-1)/2} d^{1/2}/\delta + 1)$$

657 $\qquad\qquad\qquad\qquad\qquad\qquad\qquad\qquad\qquad\qquad\qquad\qquad\qquad\qquad\qquad\qquad\qquad$ □

## D.3 Covering Number of $p$-Norm Constrained Linear Combination

659 **Lemma 6.** $\log \mathcal{N}(\mathcal{G}, \delta) \lesssim k \log(1/\delta)$ *for some finite* $c_3$, *and for any* $g \in \mathcal{G}, |a| \leq 1$, *we have*
660 $ag \in \mathcal{G}$. *The covering number of* $\mathcal{F} = \left\{\sum_{i=1}^{M} a_i g_i \middle| g_i \in \mathcal{G}, \|a\|_p^p \leq P, 0 < p < 1\right\}$ *for any* $P > 0$
661 *satisfies*

$$\log \mathcal{N}(\mathcal{F}, \epsilon) \lesssim k P^{\frac{1}{1-p}} (\delta/c_3)^{-\frac{p}{1-p}} \log(c_3 P/\delta)$$

662 *up to a double logarithmic factor.*

663 *Proof.* Let $\epsilon$ be a positive constant. Without the loss of generality, we can sort the coefficients in
664 descending order in terms of their absolute values. There exists a positive integer $\mathcal{M}$ (as a function
665 of $\epsilon$), such that $|a_i| \geq \epsilon$ for $i \leq \mathcal{M}$, and $|a_i| < \epsilon$ for $i > \mathcal{M}$.

666 By definition, $\mathcal{M}\epsilon^p \leq \sum_{i=1}^{\mathcal{M}} |a_i|^p \leq P$ so $\mathcal{M} \leq P/\epsilon^p$, and $|a_i|^p \leq P, |a_i| \leq P^{1/p}$ for all $i$.
667 Furthermore,

$$\sum_{i>m} |a_i| = \sum_{i>\mathcal{M}} |a_i|^p |a_i|^{1-p} < \sum_{i>\mathcal{M}} |a_i|^p \epsilon^{1-p} \leq P\epsilon^{1-p}$$

668 Let $\tilde{g}_i = \arg\min_{g \in \tilde{\mathcal{G}}} \|g - \frac{a_i}{P^{1/p}} g_i\|_\infty$ where $\tilde{\mathcal{G}}$ is the $\delta'$-convering set of $\mathcal{G}$. By definition of the
669 covering set,

$$\left\|\sum_{i=1}^{M} a_i g_i(x) - \sum_{i=1}^{\mathcal{M}} P^{1/p} \tilde{g}_i(x)\right\|_\infty \leq \left\|\sum_{i=1}^{\mathcal{M}} (a_i g_i(x) - P^{1/p} \tilde{g}_i(x))\right\|_\infty + \left\|\sum_{i=\mathcal{M}+1}^{M} a_i g_i(x)\right\|_\infty$$
$$\leq \mathcal{M} P^{1/p} \delta' + c_3 P \epsilon^{1-p}.$$
$$\tag{17}$$

Choosing

$$\epsilon = (\delta/2c_3 P)^{\frac{1}{1-p}}, \delta' \eqsim P^{-\frac{1}{p(1-p)}}(\delta/2c_3)^{\frac{1}{1-p}}/2, \tag{18}$$

we have $\mathcal{M} \leq P^{\frac{1}{1-p}}(\delta/2c_3)^{-\frac{p}{1-p}}, \mathcal{M}P^{1/p}\delta' \leq \delta/2, c_3 P\epsilon^{1-p} \leq \delta/2$, so (17) $\leq \delta$. One can compute the covering number of $\mathcal{F}$ by

$$\log \mathcal{N}(\mathcal{F}, \delta) \leq \mathcal{M} \log \mathcal{N}(\mathcal{G}, \delta') \lesssim k\mathcal{M} \log(1/\delta') \tag{19}$$

Taking (18) into (19) finishes the proof. $\qquad\square$

# E  Proof of Approximation Error

## E.1  Approximation of Neural Networks to B-spline Basis Functions

**Proposition 7.** *Let $M_{m,k,s}$ be the B-spline of order $m$ with scale $2^{-k}$ in each dimension and position $s \in \mathbb{R}^d$: $M_{m,k,s}(\boldsymbol{x}) := M_m(2^k(\boldsymbol{x} - \boldsymbol{s}))$, $M_m$ is defined in (11). There exists a parallel neural network that has the structure and satisfy the constraint in Proposition 4 for $d$-dimensional input and one output, containing $M = O(m^d)$ subnetworks, each of which has width $w = O(d)$ and depth $L = O(\log(c(m,d)/\epsilon))$ for some constant $w, c$ that depends only on $m$ and $d$, denoted as $\tilde{M}_m(\boldsymbol{x}), \boldsymbol{x} \in \mathbb{R}^d$, such that*

- $|\tilde{M}_{m,k,s}(\boldsymbol{x}) - M_{m,k,s}(\boldsymbol{x})| \leq \epsilon$, *if* $0 \leq 2^k(x_i - s_i) \leq m + 1, \forall i \in [d]$,
- $\tilde{M}_{m,k,s}(\boldsymbol{x}) = 0$, *otherwise*.
- *The weights in the last layer satisfy* $\|a\|_{2/L}^{2/L} \lesssim 2^k m^d e^{2md/L}$.

We follow the method developed in Yarotsky [49], Suzuki [39], while putting our attention on bounding the Frobenius norm of the weights.

**Lemma 12** (Yarotsky [49, Proposition 3]). *: There exists a neural network with two-dimensional input and one output $f_\times(x, y)$, with constant width and depth $O(\log(1/\delta))$, and the weight in each layer is bounded by a global constant $c_1$, such that*

- $|f_\times(x, y) - xy| \leq \delta, \forall\, 0 \leq x, y \leq 1$,
- $f_\times(x, y) = 0, \forall\, x = 0$ *or* $y = 0$.

We first prove a special case of Proposition 7 on the unscaled, unshifted B-spline basis function by fixing $k = 0, \boldsymbol{s} = 0$:

**Proposition 13.** *There exists a parallel neural network that has the structure and satisfy the constraint in Proposition 4 for $d$-dimensional input and one output, containing $M = \lceil (m+1)/2 \rceil^d = O(m^d)$ subnetworks, each of which has width $w = O(d)$ and depth $L = O(\log(c(m,d)/\epsilon))$ for some constant $w, c$ that depends only on $m$ and $d$, denoted as $\tilde{M}_m(\boldsymbol{x}), \boldsymbol{x} \in \mathbb{R}^d$, such that*

- $|\tilde{M}_m(\boldsymbol{x}) - M_m(\boldsymbol{x})| \leq \epsilon$, *if* $0 \leq x_i \leq m + 1, \forall i \in [d]$, *while $M_m(\cdot)$ denote $m$-th order B-spline basis function, and $c$ only depends on $m$ and $d$.*
- $\tilde{M}_m(\boldsymbol{x}) = 0$, *if* $x_i \leq 0$ *or* $x_i \geq m + 1$ *for any* $i \in [d]$.
- *The weights in the last layer satisfy* $\|a\|_{2/L}^{2/L} \lesssim m^d e^{2md/L}$.

*Proof.* We first show that one can use a neural network with constant width $w_0$, depth $L \eqsim \log(m/\epsilon_1)$ and bounded norm $\|W^{(1)}\|_F \leq O(\sqrt{d}), \|W^{(\ell)}\|_F \leq O(\sqrt{w}), \forall \ell = 2, \ldots, L$ to approximate truncated power basis function up to accuracy $\epsilon_1$ in the range $[0, 1]$. Let $m = \sum_{i=0}^{\lceil \log_2 m \rceil} m_i 2^i, m_i \in \{0, 1\}$ be the binary digits of $m$, and define $\bar{m}_j = \sum_{j=0}^{i} m_i, \gamma = \lceil \log_2 m \rceil$,

then for any $x$

$$x_+^m = x_+^{\bar{m}_\gamma} \times \left(x_+^{2^\gamma}\right)^{m_\gamma}$$

$$[x_+^{\bar{m}_\gamma}, x_+^{2^\gamma}] = [x_+^{\bar{m}_{\gamma-1}} \times \left(x_+^{2^{\gamma-1}}\right)^{m_{\gamma-1}}, x_+^{2^{\gamma-1}} \times x_+^{2^{\gamma-1}}]$$

$$\ldots \tag{20}$$

$$[x_+^{\bar{m}_2}, x_+^4] = [x_+^{\bar{m}_1} \times \left(x_+^2\right)^{m_1}, x_+^2 \times x_+^2]$$

$$[x_+^{\bar{m}_1}, x_+^2] = [x_+^{\bar{m}_0} \times x_+^{m_0}, x_+ \times x_+]$$

Notice that each line of equation only depends on the line immediately below. Replacing the multiply operator $\times$ with the neural network approximation shown in Lemma 12 demonstrates the architecture of such neural network approximation. For any $x, y \in [0,1]$, let $|f_\times(x, y) - xy| \leq \delta, |x - \tilde{x}| \leq \delta_1, |y - \delta y| \leq \delta_2$, then $|f_\times(\tilde{x}, \tilde{y}) - xy| \leq \delta_1 + \delta_2 + \delta$. Taking this into (20) shows that $\epsilon_1 \approx 2^\gamma \delta \approx m\delta$, where $\epsilon_1$ is the upper bound on the approximate error to truncated power basis of order $m$ and $\delta$ is the approximation error to a single multiply operator as in Lemma 12.

A univariate B-spline basis can be expressed using truncated power basis, and observing that it is symmetric around $(m+1)/2$:

$$M_m(x) = \frac{1}{m!} \sum_{j=1}^{m+1} (-1)^j \binom{m+1}{j} (x-j)_+^m$$

$$= \frac{1}{m!} \sum_{j=1}^{\lceil (m+1)/2 \rceil} (-1)^j \binom{m+1}{j} (\min(x, m+1-x) - j)_+^m$$

$$= \frac{((m+1)/2)^m}{m!} \sum_{j=1}^{\lceil (m+1)/2 \rceil} (-1)^j \binom{m+1}{j} \left(\frac{\min(x, m+1-x) - j}{(m+1)/2}\right)_+^m,$$

A multivariate ($d$-dimensional) B-spline basis function can be expressed as the product of truncated power basis functions and thus can be decomposed as

$$M_m(\boldsymbol{x}) = \prod_{i=1}^{d} M_m(x_i)$$

$$= \frac{((m+1)/2)^{md}}{(m!)^d} \prod_{i=1}^{d} \left( \sum_{j=1}^{\lceil (m+1)/2 \rceil} (-1)^j \binom{m+1}{j} \left(\frac{\min(x_i, m+1-x) - j}{(m+1)/2}\right)_+^m \right) \tag{21}$$

$$= \frac{((m+1)/2)^{md}}{(m!)^d} \sum_{j_1,\ldots,j_d=1}^{\lceil (m+1)/2 \rceil} \prod_{i=1}^{d} (-1)^{j_i} \binom{m+1}{j_i} \left(\frac{\min(x, m+1-x) - j_i}{(m+1)/2}\right)_+^m$$

Using Lemma 12, one can construct a parallel neural network containing $M = \lceil (m+1)/2 \rceil^d = O(m^d)$ subnetworks, and each subnetwork corresponds to one polynomial term in (21). Using the results above, the approximation of this constructed neural network can be bounded by

$$\frac{((m+1)/2)^{md}}{(m!)^d} \sum_{j_1,\ldots,j_d=1}^{\lceil (m+1)/2 \rceil} \prod_{i=1}^{d} (-1)^{j_i} \binom{m+1}{j_i} \epsilon_1 \lesssim e^{md} \epsilon_1$$

where we applied Stirling's approximation and $\delta$ and $\epsilon_1$ has the same definition as above. Choosing $\delta = \frac{\epsilon}{d(e^{2m}\sqrt{m+1})}$, and recall $\epsilon_1 \approx m\delta$ proves the approximation error.

To bound the norm of the factors $\|a\|_{2/L}^{2/L}$, first observe that

$$|a_{j_1,\ldots,j_d}| = \frac{((m+1)/2)^{md}}{(m!)^d} \frac{1}{(m+1)/2} \prod_{i=1}^{d} \binom{m+1}{j_i}$$

$$\leq \frac{((m+1)/2)^{md}}{(m!)^d} \frac{2^{md}}{(m+1)/2} = O(e^{md})$$

where the first inequality is from $\binom{m+1}{j_i} \leq 2^{m+1}$, the last equality is from Stirling's appropximation. Finally,

$$\|a\|_{2/L}^{2/L} \leq m^d \max_j |a_j|^{2/L} \lesssim m^d e^{2md/L}$$

which finishes the proof. □

The proof of the Proposition 7 for general $k, s$ follows by appending one more layer in the front, as we show below.

*Proof of Proposition 7.* Using the neural network proposed in Proposition 13, one can construct a neural network for appropximating $M_{m,k,s}$ by adding one layer before the first layer:

$$\sigma(2^k \mathbf{I}_d \boldsymbol{x} - 2^k \boldsymbol{s})$$

The unused neurons in the first hidden layer is zero padded. The Frobenius norm of the weight is $2^k \|\mathbf{I}_d\|_F = 2^k \sqrt{d}$. Following the proof of Proposition 4, rescaling the weight in this layer by $2^{-k}$, and the weight matrix in the last layer by $2^k$, and scaling the bias properly, one can verify that this neural network satisfy the statement. □

## E.2 Sparse approximation of Besov functions using B-spline wavelets

**Proposition 8.** *Let $\alpha - d/p > 1, r > 0$. For any function in Besov space $f_0 \in B_{p,q}^\alpha$ and any positive integer $\bar{M}$, there is an $\bar{M}$-sparse approximation using B-spline basis of order $m$ satisfying $0 < \alpha < \min(m, m - 1 + 1/p)$: $\check{f}_{\bar{M}} = \sum_{i=1}^{\bar{M}} a_{k_i, s_i} M_{m, k_i, s_i}$ for any positive integer $\bar{M}$ such that the approximation error is bounded as $\|\check{f}_{\bar{M}} - f_0\|_r \lesssim \bar{M}^{-\alpha/d} \|f_0\|_{B_{p,q}^\alpha}$, and the coefficients satisfy*

$$\|\{2^{k_i} a_{k_i, s_i}\}_{k_i, s_i}\|_p \lesssim \|f_0\|_{B_{p,q}^\alpha}.$$

The proof is divided into three steps:

1. Bound the 0-norm and the 1-norm of the coefficients of B-spline basis in order to approximate an arbitrary function in Besov space up to any $\epsilon > 0$.
2. Bound $p$-norm of the coefficients of B-spline basis functions where $0 < p < 1$ using the results above .
3. Add the approximation to neural network to B-spline basis computed in Section 4.3.1 into Step 2.

*Proof.* Dũng [11, Theorem 3.1] Suzuki [39, Lemma 2] proposed an adaptive sampling recovery method that approximates a function in Besov space. The method is divided into two cases: when $p \geq r$, and when $p < r$.

When $p \geq r$, there exists a sequence of scalars $\lambda_{\boldsymbol{j}}, \boldsymbol{j} \in P^d(\mu), P_d(\mu) := \{\boldsymbol{j} \in \mathbb{Z}^d : |j_i| \leq \mu, \forall i \in d\}$ for some positive $\mu$, for arbitrary positive integer $\bar{k}$, the linear operator

$$Q_{\bar{k}}(f, \boldsymbol{x}) = \sum_{\boldsymbol{s} \in J(\bar{k}, m, d)} a_{\bar{k}, \boldsymbol{s}}(f) M_{\bar{k}, \boldsymbol{s}}(\boldsymbol{x}), \quad a_{\bar{k}, \boldsymbol{s}}(f) = \sum_{\boldsymbol{j} \in \mathbb{Z}^d, P^d(\mu)} \lambda_{\boldsymbol{j}} \bar{f}(\boldsymbol{s} + 2^{-\bar{k}} \boldsymbol{j})$$

has bounded approximation error

$$\|f - Q_{\bar{k}}(f, x)\|_r \leq C 2^{-\alpha \bar{k}} \|f\|_{B_{p,q}^\alpha},$$

where $\bar{f}$ is the extrapolation of $f$, $J(\bar{k}, m, d) := \{\boldsymbol{s} : 2^{\bar{k}} \boldsymbol{s} \in \mathbb{Z}^d, -m/2 \leq 2^{\bar{k}} s_i \leq 2^{\bar{k}} + m/2, \forall i \in [d]\}$. See Dũng [11, 2.6-2.7] for the detail of the extrapolation as well as references for options of sequence $\lambda_{\boldsymbol{j}}$.

Furthermore, $Q_{\bar{k}}(f) \in B_{p,q}^\alpha$ so it can be decomposed in the form (10) with $M = \sum_{k=0}^{\bar{k}} (2^k + m - 1)^d \lesssim 2^{\bar{k}d}$ components and $\|\{\tilde{c}_{k,s}\}_{k,s}\| \lesssim \|Q_{\bar{k}}(f)\|_{B_{p,q}^\alpha} \lesssim \|f\|_{B_{p,q}^\alpha}$ where $\tilde{c}_{k,s}$ is the coefficients of the decomposition of $Q_{\bar{k}}(f)$. Choosing $\bar{k} \approx \log_2 M/d$ leads to the desired approximation error.

On the other hand, when $p < r$, there exists a greedy algorithm that constructs

$$G(f) = Q_{\bar{k}}(f) + \sum_{k=\bar{k}+1}^{k^*} \sum_{j=1}^{n_k} c_{k,\boldsymbol{s}_j}(f) M_{k,\boldsymbol{s}_j}$$

where $\bar{k} \asymp \log_2(M), k^* = [\epsilon^{-1} \log(\lambda M)] + \bar{k} + 1, n_k = [\lambda M 2^{-\epsilon(k-\bar{k})}]$ for some $0 < \epsilon < \alpha/\delta - 1, \delta = d(1/p - 1/r), \lambda > 0$, such that

$$\|f - G(f)\|_r \leq \bar{M}^{-\alpha/d}\|f\|_{B_{p,q}^\alpha}$$

and

$$\sum_{k=0}^{\bar{k}} (2^k + m - 1)^d + \sum_{k=\bar{k}+1}^{k^*} n_k \leq \bar{M}.$$

See Dũng [11, Theorem 3.1] for the detail.

Finally, since $\alpha - d/p > 1$,

$$
\begin{aligned}
\|\{2^{k_i} c_{k_i,\boldsymbol{s}_i}\}_{k_i,\boldsymbol{s}_i}\|_p &\leq \sum_{k=0}^{\bar{k}} 2^k \|\{c_{k_i,\boldsymbol{s}_i}\}_{\boldsymbol{s}_i}\|_p \\
&= \sum_{k=0}^{\bar{k}} 2^{(1-(\alpha-d/p))k}(2^{(\alpha-d/p)k}\|\{c_{k_i,\boldsymbol{s}_i}\}_{\boldsymbol{s}_i}\|_p) \\
&\lesssim \sum_{k=0}^{\bar{k}} 2^{(1-(\alpha-d/p))k}\|f\|_{B_{p,q}^\alpha} \\
&\asymp \|f\|_{B_{p,q}^\alpha}
\end{aligned}
\tag{22}
$$

where the first line is because for arbitrary vectors $\boldsymbol{a}_i, i \in [n], \|\sum_{i=1}^n \boldsymbol{a}_i\|_p \leq \sum_{i=1}^n \|\boldsymbol{a}_i\|_p$, the third line is because the sequence norm of B-spline decomposition is equivalent to the norm in Besov space (see Section C.1) . $\qquad\square$

Note that when $\alpha - d/p = 1$, the sequence norm (22) is bounded (up to a factor of constant) by $k^*\|f\|_{B_{p,q}^\alpha}$, which can be proven by following (22) except the last line. This adds a logarithmic term with respect to $\bar{M}$ compared with the result in Proposition 8. This will add a logarithmic factor to the MSE. We will not focus on this case in this paper of simplicity.

### E.3 Sparse approximation of Besov functions using Parallel Neural Networks

**Theorem 9.** *Under the same condition as Proposition 8, for any positive integer $\bar{M}$, any function in Besov space $f_0 \in B_{p,q}^\alpha$ can be approximated by a parallel neural network with no less than $O(m^d \bar{M})$ number of subnetworks satisfying:*

1. *Each subnetwork has width $w = O(d)$ and depth L.*

2. *The weights in each layer satisfy $\|\bar{\mathbf{W}}_k^{(\ell)}\|_F \leq O(\sqrt{w})$ except the first layer $\|\bar{\mathbf{W}}_k^{(1)}\|_F \leq O(\sqrt{d})$,*

3. *The scaling factors have bounded $2/L$-norm: $\|\{a_j\}\|_{2/L}^{2/L} \lesssim m^d e^{2md/L} \bar{M}^{1-2/(pL)}$.*

4. *The approximation error is bounded by*

$$\|\tilde{f} - f_0\|_r \leq (c_4 \bar{M}^{-\alpha/d} + c_5 e^{-c_6 L})\|f\|_{B_{p,q}^\alpha}$$

*where $c_4, c_5, c_6$ are constants that depend only on $m, d$ and $p$.*

We first prove the following lemma.

**Lemma 14.** *For any $a \in \mathbb{R}^{\bar{M}}, 0 < p' < p$, it holds that:*

$$\|a\|_{p'}^{p'} \leq \bar{M}^{1-p'/p}\|a\|_p^{p'}.$$

*Proof.*

$$\sum_i |a_i|^{p'} = \langle \mathbf{1}, |\boldsymbol{a}|^{p'} \rangle \leq \left( \sum_i 1 \right)^{1-\frac{p'}{p}} \left( \sum_i (|a_i|^{p'})^{\frac{p}{p'}} \right)^{\frac{p'}{p}} = \bar{M}^{1-\frac{p'}{p}} \|a\|_p^{p'}$$

The first inequality uses a Holder's inequality with conjugate pair $\frac{p}{p'}$ and $1/(1-\frac{p'}{p})$. $\qquad\square$

*Proof of Theorem 9.* Using Proposition 8, one can construct $\bar{M}$ number of PNN each $O(m^d)$ sub-networks according to Proposition 7, and in each PNN, such that each PNN represents one B-spline basis function.The weights in the last layer of each PNN is scaled to match the coefficients in Proposition 8. Taking $p'$ in Lemma 14 as $2/L$ and combining with Proposition 7 finishes the proof. $\qquad\square$

## F  Proof of the Main Theorem

**Theorem 1 extended form.** *For any fixed $\alpha - d/p > 1, q \geq 1, L \geq 3$, for any $f_0 \in B_{p,q}^\alpha$, given an $L$-layer parallel neural network satisfying*

- *The width of each subnetwork is fixed and large enough: $w \gtrsim d$. See Theorem 9 for the detail.*

- *The number of subnetworks is large enough: $M \gtrsim m^d n^{\frac{1-2/L}{2\alpha/d+1-2/(pL)}}$ where $m = \lceil \alpha - 1 \rceil$.*

*With proper choice of the parameter of weight decay $\lambda$, the solution $\hat{f}$ parameterized by (2) satisfies*

$$\mathrm{MSE}(\hat{f}) = \tilde{O}\left( \left( \frac{w^{4-4/L} L^{2-4/L}}{n^{1-2/L}} \right)^{\frac{2\alpha/d}{2\alpha/d+1-2/(pL)}} + e^{-c_6 L} \right)$$

*where $\tilde{O}$ shows the scale up to a logarithmic factor, and $c_6$ is the constant defined in Theorem 9.*

*Proof.* First recall the relationship between covering number (entropy) and estimation error:

**Proposition 15.** *Let $\mathcal{F} \subseteq \{\mathbb{R}^d \to [-F, F]\}$ be a set of functions. Assume that $\mathcal{F}$ can be decomposed into two orthogonal spaces $\mathcal{F} = \mathcal{F}_\| \times \mathcal{F}_\perp$ where $\mathcal{F}_\perp$ is an affine space with dimension of $N$. Let $f_0 \in \{\mathbb{R}^d \to [-F, F]\}$ be the target function and $\hat{f}$ be the least squares estimator in $\mathcal{F}$:*

$$\hat{f} = \arg\min_{f \in \mathcal{F}} \sum_{i=1}^n (y_i - f(x_i))^2, y_i = f_0(x_i) + \epsilon_i, \epsilon_i \sim \mathcal{N}(0, \sigma^2) i.i.d.,$$

*then it holds that*

$$\mathrm{MSE}(\hat{f}) \leq \tilde{O}\left( \arg\min_{f \in \mathcal{F}} \mathrm{MSE}(f) + \frac{N + \log \mathcal{N}(\mathcal{F}_\|, \delta) + 2}{n} + (F + \sigma)\delta \right).$$

The proof of Proposition 15 is defered to the section below. We choose $\mathcal{F}$ as the set of functions that can be represented by a parallel neural network as stated, the (null) space $\mathcal{F}_\perp = \{f : f(\boldsymbol{x}) = constant\}$ be the set of functions with constant output, which has dimension 1. This space captures the bias in the last layer, while the other parameters contributes to the projection in $\mathcal{F}_\|$. See Section D.2 for how we handle the bias in the other layers. One can find that $\mathcal{F}_\|$ is the set of functions that can be represented by a parallel neural network as stated, and further satisfy $\sum_{i=1}^n f(\boldsymbol{x}_i) = 0$. Because $\mathcal{F}_\| \subseteq \mathcal{F}, \mathcal{N}(\mathcal{F}_\|, \delta) \leq \mathcal{N}(\mathcal{F}, \delta)$ for all $\delta > 0$, and the latter is studied in Theorem 5.

In Theorem 1, the width of each subnetwork is no less than what is required in Theorem 9, while the depth and norm constraint are the same, so the approximation error is no more that that in Theorem 9. Choosing $r = 2, p = 2/L$, and taking Theorem 5 and Theorem 9 into this Proposition 15, one gets

$$\mathrm{MSE}(\hat{f}) \lesssim \bar{M}^{-2\alpha/d} + \frac{w^{2+2/(1-2/L)} L^2}{n} \bar{M}^{\frac{1-2/(pL)}{1-2/L}} \delta^{-\frac{2/L}{1-2/L}} \left( \log(\bar{M}/\delta) + 3 \right) + \delta,$$

where $\|f\|_{B_{p,q}^\alpha}, m$ and $d$ taken as constants. The stated MSE is obtained by choosing

$$\delta \approx \frac{w^{4-4/L}L^{2-4/L}\bar{M}^{1-2/(pL)}}{n^{1-2/L}}, \quad \bar{M} \approx \left(\frac{n^{1-2/L}}{w^{4-4/L}L^{2-4/L}}\right)^{\frac{1}{2\alpha/d+1-2/(pL)}}$$

Note that there exists a weight decay parameter $\lambda'$ such that the $(2/L)$-norm of the coefficients of the parallel neural network satisfy that $\|\{a_j\}\|_{2/L}^{2/L} = m^d e^{2md/L}\|\{\tilde{a}_{j,\bar{M}}\}\|_{2/L}^{2/L}$ where $\{\tilde{a}_{j,\bar{M}}\}$ is the coefficient of the particular $\bar{M}$-sparse approximation, although $\{a_j\}$ is not necessarily $\bar{M}$ sparse. Empirically, one only need to guarantee that during initialization, the number of subnetworks $M \geq \bar{M}$ such that the $\bar{M}$-sparse approximation is feasible, thus the approximation error bound from Theorem 9 can be applied. Theorem 9 also says that $\|\{a_j\}\|_{2/L}^{2/L} = m^d e^{2md/L}\|\{\tilde{a}_{j,\bar{M}}\}\|_{2/L}^{2/L} \lesssim \bar{M}^{1-2/pL}$, thus we can apply the covering number bound from Theorem 5 with $P' = \bar{M}^{1-2/pL}$. Finally, if $\lambda$ is optimally chosen, then it achieves a smaller MSE than this particular $\lambda'$, which has been proven to be no more than $O(\bar{M}^{-\alpha/d})$ and completes the proof.

$\square$

*Proof of Proposition 15.* For any function $f \in \mathcal{F}$, define $f_\perp = \arg\min_{h \in \mathcal{F}_\perp} \sum_{i=1}^n (f(\boldsymbol{x}_i) - h(\boldsymbol{x}_i))^2$ be the projection of $f$ to $\mathcal{F}_\perp$, and define $f_\| = f - f_\perp$ be the projection to the orthogonal complement. Note that $f_\|$ is not necessarily in $\mathcal{F}_\|$. However, if $f \in \mathcal{F}$, then $f_\| \in \mathcal{F}_\|$. $y_{i\perp}$ and $y_{i\|}$ are defined by creating a function $f_y$ such that $f_y(\boldsymbol{x}_i) = y_i, \forall i$, e.g. via interpolation. Because $\mathcal{F}_\|$ and $\mathcal{F}_\perp$ are orthononal, the empirical loss and population loss can be decomposed in the same way:

$$L_\|(f) = \frac{1}{n}\sum_{i=1}^n (f_\|(\boldsymbol{x}) - f_{0\|}(\boldsymbol{x}))^2 + \frac{n-N}{n}\sigma^2, \qquad L_\perp(f) = \frac{1}{n}\sum_{i=1}^n (f_\perp(\boldsymbol{x}) - f_{0\perp}(\boldsymbol{x}))^2 + \frac{N}{n}\sigma^2,$$

$$\hat{L}_\|(f) = \frac{1}{n}\sum_{i=1}^n (f_\|(\boldsymbol{x}) - y_{i\|})^2, \qquad \hat{L}_\perp(f) = \frac{1}{n}\sum_{i=1}^n (f_\perp(\boldsymbol{x}) - y_{i\perp}(\boldsymbol{x}))^2,$$

$$MSE_\|(f) = \mathbb{E}_{\mathcal{D}}\left[\frac{1}{n}\sum_{i=1}^n (f_\|(\boldsymbol{x}) - f_{0\|}(\boldsymbol{x}))^2\right], \qquad MSE_\perp(f) = \mathbb{E}_{\mathcal{D}}\left[\frac{1}{n}\sum_{i=1}^n (f_\perp(\boldsymbol{x}) - f_{0\perp}(\boldsymbol{x}))^2\right],$$

such that $L(f) = L_\|(f) + L_\perp(f), \hat{L}(f) = \hat{L}_\|(f) + \hat{L}_\perp(f)$. This can be verified by decomposing $\hat{f}, f_0$ and $y$ into two orthogonal components as shown above, and observing that $\sum_{i=1}^n f_{1\perp}(\boldsymbol{x}_i)f_{2\|}(\boldsymbol{x}_i) = 0, \forall f_1, f_2$.

**First prove the following claim**

**Claim 16.** *Assume that $\hat{f} = \arg\min_{f \in \mathcal{F}} \hat{L}(f)$ is the empirical risk minimizer. Then $\hat{f}_\perp = \arg\min_{f \in \mathcal{F}_\perp} \hat{L}_\perp(f), \hat{f}_\| = \arg\min_{f \in \mathcal{F}_\|} \hat{L}_\|(f)$, where $\hat{f}_\perp$ is the projections of $\hat{f}$ in $\mathcal{F}_\perp$, and $\hat{f}_\| = \hat{f} - \hat{f}_\perp$ respectively.*

*Proof.* Since $\hat{f} \in \mathcal{F}$, by definition $\hat{f}_\| \in \mathcal{F}_\|$. Assume that there exist $\hat{f}'_\perp, \hat{f}'_\|$, and either $\hat{L}_\perp(\hat{f}'_\perp) < \hat{L}_\perp(\hat{f}_\perp)$, or $\hat{L}_\|(\hat{f}'_\|) < \hat{L}_\|(\hat{f}_\|)$. Then

$$\hat{L}(\hat{f}') = \hat{L}(\hat{f}'_\perp + \hat{f}'_\|) = \hat{L}_\|(\hat{f}'_\perp + \hat{f}'_\|) + \hat{L}_\perp(\hat{f}'_\perp + \hat{f}'_\|) = \hat{L}_\|(\hat{f}'_\|) + \hat{L}_\perp(\hat{f}'_\perp)$$
$$< \hat{L}_\|(\hat{f}_\|) + \hat{L}_\perp(\hat{f}_\perp) = \hat{L}_\|(\hat{f}_\perp + \hat{f}_\|) + \hat{L}_\perp(\hat{f}_\perp + \hat{f}_\|) = \hat{L}(\hat{f})$$

which shows that $\hat{f}$ is not the minimizer of $\hat{L}(f)$ and violates the assumption.

$\square$

**Then we bound** $MSE_\perp(f)$**.** We convert this part into a finite dimension least square problem:

$$\hat{f}_\perp = \arg\min_{f\in\mathcal{F}_\perp} \hat{L}_\perp(f)$$

$$= \arg\min_{f\in\mathcal{F}_\perp} \frac{1}{n}\sum_{i=1}^{n}(f(\boldsymbol{x}_i) - f_{0\perp}(\boldsymbol{x}_i) - \epsilon_{i\perp})^2$$

$$= \arg\min_{f\in\mathcal{F}_\perp} \frac{1}{n}\sum_{i=1}^{n}(f(\boldsymbol{x}_i) - f_{0\perp}(\boldsymbol{x}_i) - \epsilon_{i\perp})^2 + \epsilon_{i\parallel}^2$$

$$= \arg\min_{f\in\mathcal{F}_\perp} \frac{1}{n}\sum_{i=1}^{n}(f(\boldsymbol{x}_i) - f_{0\perp}(\boldsymbol{x}_i) - \epsilon_{i\perp} - \epsilon_{i\parallel})^2$$

$$= \arg\min_{f\in\mathcal{F}_\perp} \frac{1}{n}\sum_{i=1}^{n}(f(\boldsymbol{x}_i) - f_{0\perp}(\boldsymbol{x}_i) - \epsilon_i)^2$$

The forth line comes from our assumption that $\mathcal{F}_\perp$ is orthogonal to $\mathcal{F}_\parallel$, so $\forall f \in \mathcal{F}_\perp, f + f_{0\perp} + \epsilon_\perp$ is orthogonal to $\epsilon_\parallel$.

Let the basis function of $\mathcal{F}_\perp$ be $h_1, h_2, \ldots, h_N$, the above problem can be reparameterized as

$$\arg\min_{\boldsymbol{\theta}\in\mathbb{R}^N} \frac{1}{n}\|\mathbf{X}\boldsymbol{\theta} - \boldsymbol{y}\|^2$$

where $\mathbf{X} \in \mathbb{R}^{n\times N} : X_i = h_j(\boldsymbol{x}_i), \boldsymbol{y} = \boldsymbol{y}_{0\perp} + \boldsymbol{\epsilon}, \boldsymbol{y}_{0\perp} = [f_{0\perp}(x_1), \ldots, f_{0\perp}(x_n)], \boldsymbol{\epsilon} = [\epsilon_1, \ldots, \epsilon_n]$. This problem has a closed-form solution

$$\boldsymbol{\theta} = (\mathbf{X}^T\mathbf{X})^{-1}\mathbf{X}^T\boldsymbol{y}$$

Observe that $f_{0\perp} \in \mathcal{F}_\perp$, let $\boldsymbol{y}_{0\perp} = \mathbf{X}\boldsymbol{\theta}^*$,The MSE of this problem can be computed by

$$L(\hat{f}_\perp) = \frac{1}{n}\|\mathbf{X}\boldsymbol{\theta} - \boldsymbol{y}_{0\perp}\|^2 = \frac{1}{n}\|\mathbf{X}(\mathbf{X}^T\mathbf{X})^{-1}\mathbf{X}^T(\mathbf{X}\boldsymbol{\theta}^* + \boldsymbol{\epsilon}) - \mathbf{X}\boldsymbol{\theta}^*\|^2$$

$$= \frac{1}{n}\|\mathbf{X}(\mathbf{X}^T\mathbf{X})^{-1}\mathbf{X}^T\boldsymbol{\epsilon}\|^2$$

Observing that $\Pi := \mathbf{X}(\mathbf{X}^T\mathbf{X})^{-1}\mathbf{X}^T$ is an idempotent and independent projection whose rank is $N$, and that $\mathbb{E}[\boldsymbol{\epsilon}\boldsymbol{\epsilon}^T] = \sigma^2\mathbf{I}$, we get

$$\mathrm{MSE}_\perp(\hat{f}_\perp) = \mathbb{E}[L(\hat{f}_\perp)] = \frac{1}{n}\|\Pi\boldsymbol{\epsilon}\|^2 = \frac{1}{n}\mathrm{tr}(\Pi\boldsymbol{\epsilon}\boldsymbol{\epsilon}^T) = \frac{\sigma^2}{n}\mathrm{tr}(\Pi)$$

which concludes that

$$\mathrm{MSE}_\perp(\hat{f}) = O\left(\frac{N}{n}\sigma^2\right). \tag{23}$$

See also [19, Proposition 1].

**Next we study** $\mathrm{MSE}_\parallel(\hat{f})$**.** Denote $\tilde{\sigma}_\parallel^2 = \frac{1}{n}\sum_{i=1}^{n}\epsilon_{i\parallel}^2, E = \max_i |\epsilon_i|$. Using Jensen's inequality and union bound, we have

$$\exp(t\mathbb{E}[E]) \leq \mathbb{E}[\exp(tE)] = \mathbb{E}[\max\exp(t|\epsilon_i|)] \leq \sum_{i=1}^{n}\mathbb{E}[\exp(t|\epsilon_i|)] \leq 2n\exp(t^2\sigma^2/2)$$

Taking expectation over both sides, we get

$$\mathbb{E}[E] \leq \frac{\log 2n}{t} + \frac{t\sigma^2}{2}$$

maximizing the right hand side over $t$ yields

$$\mathbb{E}[E] \leq \sigma\sqrt{2\log 2n}.$$

Let $\tilde{\mathcal{F}}_\|$ be the covering set of $\mathcal{F}_\| = \{f_\| : f \in \mathcal{F}\}$. For any $\tilde{f}_\| \in \tilde{\mathcal{F}}_\|$,

$$
\begin{aligned}
L_\|(f_j) - \hat{L}_\|(f_j) &= \frac{1}{n}\sum_{i=1}^n (f_{j\|}(\boldsymbol{x}_i) - f_{0\|}(\boldsymbol{x}_i))^2 - \frac{1}{n}\sum_{i=1}^n (\tilde{f}_\|(\boldsymbol{x}_i) - y_{i\|})^2 + \frac{n-N}{n}\sigma^2 \\
&= \frac{1}{n}\sum_{i=1}^n \epsilon_{i\|}(2\tilde{f}_\|(\boldsymbol{x}_i) - f_{0\|}(\boldsymbol{x}_i) - y_{i\|}) + \frac{n-N}{n}\sigma^2 \\
&= \frac{1}{n}\sum_{i=1}^n \epsilon_i(2\tilde{f}_\|(\boldsymbol{x}_i) - f_{0\|}(\boldsymbol{x}_i) - y_{i\|}) + \frac{n-N}{n}\sigma^2 \\
&= \frac{1}{n}\sum_{i=1}^n \epsilon_i(2\tilde{f}_\|(\boldsymbol{x}_i) - 2f_{0\|}(\boldsymbol{x}_i)) + \frac{n-N}{n}\sigma^2 - \tilde{\sigma}_\|^2
\end{aligned}
$$

The first term can be bounded using Bernstein's inequality: let $h_i = \epsilon_i(f_{j\|}(\boldsymbol{x}_i) - f_{0\|}(\boldsymbol{x}_i))$, by definition $|h_i| \le 2EF$,

$$
\begin{aligned}
\mathrm{Var}[h_i] &= \mathbb{E}[\epsilon_i^2(\tilde{f}_\|(\boldsymbol{x}_i) - f_{0\|}(\boldsymbol{x}_i))^2] \\
&= (\tilde{f}_\|(\boldsymbol{x}_i) - f_{0\|}(\boldsymbol{x}_i))^2 \mathbb{E}[\epsilon_i^2] \\
&= (\tilde{f}_\|(\boldsymbol{x}_i) - f_{0\|}(\boldsymbol{x}_i))^2 \sigma^2
\end{aligned}
$$

using Bernstein's inequality, for any $\tilde{f}_\| \in \tilde{\mathcal{F}}_\|$, with probably at least $1 - \delta_p$,

$$
\begin{aligned}
\frac{1}{n}\sum_{i=1}^n \epsilon_i(2\tilde{f}_\|(\boldsymbol{x}_i) - 2f_{0\|}(\boldsymbol{x}_i)) &= \frac{2}{n}\sum_{i=1}^n h_i \\
&\le \frac{2}{n}\sqrt{2\sum_{i=1}^n (\tilde{f}_\|(\boldsymbol{x}_i) - f_{0\|}(\boldsymbol{x}_i))^2 \sigma^2 \log(1/\delta_p)} + \frac{8EF\log(1/\delta_p)}{3n} \\
&= 2\sqrt{\left(L_\|(\tilde{f}_\|) - \frac{n-N}{n}\sigma^2\right)\frac{2\sigma^2\log(1/\delta_p)}{n}} + \frac{8EF\log(1/\delta_p)}{3n} \\
&\le \epsilon\left(L_\|(\tilde{f}_\|) - \frac{n-N}{n}\sigma^2\right) + \frac{8\sigma^2\log(1/\delta_p)}{n\epsilon} + \frac{8EF\log(1/\delta_p)}{3n}
\end{aligned}
$$

the last inequality holds true for all $\epsilon > 0$. The union bound shows that with probably at least $1 - \delta$, for all $\tilde{f}_\| \in \tilde{\mathcal{F}}_\|$,

$$
\begin{aligned}
L_\|(\tilde{f}_\|) - \hat{L}_\|(\tilde{f}_\|) &\le \epsilon\left(L_\|(\tilde{f}_\|) - \frac{n-N}{n}\sigma^2\right) + \frac{8\sigma^2\log(\mathcal{N}(\mathcal{F}_\|,\delta)/\delta_p)}{n\epsilon} + \frac{8EF\log(\mathcal{N}(\mathcal{F}_\|,\delta)/\delta_p)}{3n} \\
&\quad + \frac{n-N}{n}\sigma^2 - \tilde{\sigma}_\|^2.
\end{aligned}
$$

By rearranging the terms and using the definition of $L(\tilde{f}_\|)$, we get

$$
(1-\epsilon)\left(L_\|(\tilde{f}_\|) - \frac{n-N}{n}\sigma^2\right) \le \hat{L}_\|(\tilde{f}_\|) + \frac{8\sigma^2\log(\mathcal{N}(\mathcal{F}_\|,\delta)/\delta_p)}{n\epsilon} + \frac{8EF\log(\mathcal{N}(\mathcal{F}_\|,\delta)/\delta_p)}{3n} - \tilde{\sigma}_\|^2.
$$

Taking the expectation (over $\mathcal{D}$) on both sides, and notice that $\mathbb{E}[\tilde{\sigma}_\|^2] = \frac{n-N}{n}\sigma^2$. Furthermore, for any random variable $X$, $\mathbb{E}[X] = \int_{-\infty}^\infty x \, dP(X \le x)$, we get

$$
\begin{aligned}
\max_{\tilde{f}_\| \in \tilde{\mathcal{F}}_\|} &\left((1-\epsilon)MSE_\|(\tilde{f}_\|) - \mathbb{E}[\hat{L}_\|(\tilde{f}_\|)]\right) \\
&\le \left(\frac{8\sigma^2}{n\epsilon} + \frac{8F\sigma\sqrt{2\log 2n}}{3n}\right)\left(\log\mathcal{N}(\mathcal{F}_\|,\delta) - \int_{\delta=0}^1 \log(\delta_p)d\delta_p\right) - \frac{n-N}{n}\sigma^2 \qquad (24) \\
&= \left(\frac{8\sigma^2}{n\epsilon} + \frac{8F\sigma\sqrt{2\log 2n}}{3n}\right)(\log\mathcal{N}(\mathcal{F}_\|,\delta) + 1) - \frac{n-N}{n}\sigma^2.
\end{aligned}
$$

where the integration can be computed by replacing $\delta$ with $e^x$. Though it is not integrable under Riemann integral, it is integrable under Lebesgue integration.

Similarly, let $\check{f}_\| = \arg\min_{f \in \mathcal{F}_\|} L_\|(f)$,

$$L_\|(\check{f}_\|) - \hat{L}_\|(\check{f}_\|) = \frac{1}{n}\sum_{i=1}^n \epsilon_i(2\check{f}_\|(\boldsymbol{x}_i) - 2f_{0\|}(\boldsymbol{x}_i)) + \frac{n-N}{n}\sigma^2 - \tilde{\sigma}_\|^2$$

with probably at least $1 - \delta_q$, for any $\epsilon > 0$,

$$-\frac{1}{n}\sum_{i=1}^n \epsilon_i(2\check{f}_\|(\boldsymbol{x}_i) - 2f_{0\|}(\boldsymbol{x}_i)) \le \epsilon\Big(L_\|(\check{f}_\|) - \frac{n-N}{n}\sigma^2\Big) + \frac{8\sigma^2\log(1/\delta_p)}{n\epsilon} + \frac{8EF\log(1/\delta_p)}{3n},$$

$$\hat{L}_\|(\check{f}_\|) \le (1+\epsilon)\Big(L_\|(\check{f}_\|) - \frac{n-N}{n}\sigma^2\Big) + \frac{8\sigma^2\log(1/\delta_p)}{n\epsilon} + \frac{8EF\log(1/\delta_q)}{3n} + \tilde{\sigma}_\|^2.$$

Taking the expectation on both sides,

$$\mathbb{E}[\hat{L}_\|(\check{f}_\|)] \le (1+\epsilon)\mathrm{MSE}_\|(\check{f}_\|) + \frac{8\sigma^2}{n\epsilon} + \frac{8F\sigma\sqrt{2\log 2n}}{3n} + \frac{n-N}{n}\sigma^2. \tag{25}$$

Finally, let $\hat{f}_* := \arg\min_{f \in \tilde{\mathcal{F}}_\|} \sum_{i=1}^n (\hat{f}_\|(\boldsymbol{x}_i) - f(\boldsymbol{x}_i))^2$ be the projection of $\hat{f}_\|$ in its $\delta$-covering space,

$$\begin{aligned}
\mathrm{MSE}_\|(\hat{f}_\|) &= \mathbb{E}\Big[\frac{1}{n}\sum_{i=1}^n (\hat{f}_\|(\boldsymbol{x}_i) - f_{0\|}(\boldsymbol{x}_i))^2\Big] \\
&= \mathbb{E}\Big[\frac{1}{n}\sum_{i=1}^n (\hat{f}_*(\boldsymbol{x}_i) - f_{0\|}(\boldsymbol{x}_i))^2 + \frac{1}{n}\sum_{i=1}^n (\hat{f}_\|(\boldsymbol{x}_i) - \hat{f}_*(\boldsymbol{x}_i))(\hat{f}_\|(\boldsymbol{x}_i) + \hat{f}_*(\boldsymbol{x}_i) - 2f_{0\|}(\boldsymbol{x}_i))\Big] \\
&\le \mathbb{E}\Big[\frac{1}{n}\sum_{i=1}^n (\hat{f}_*(\boldsymbol{x}_i) - f_{0\|}(\boldsymbol{x}_i))^2\Big] + 4F\delta \\
&= \mathrm{MSE}_\|(\hat{f}_*(\boldsymbol{x}_i)) + 4F\delta,
\end{aligned}$$

and similarly

$$\hat{L}_\|(\hat{f}_*) \le \hat{L}_\|(\hat{f}_\|) + (4F + 2E)\delta. \tag{26}$$

We can conclude that

$$\begin{aligned}
\mathrm{MSE}_\|(\hat{f}_\|) &\le \frac{1}{1-\epsilon}\Big(\mathbb{E}[\hat{L}_\|(\hat{f}_*)] + \Big(\frac{8\sigma^2}{n\epsilon} + \frac{8F\sigma\sqrt{2\log 2n}}{3n}\Big)(\log\mathcal{N}(\mathcal{F}_\|,\delta) + 1) - \frac{n-N}{n}\sigma^2\Big) \\
&\quad + 4F\delta \\
&\le \frac{1}{1-\epsilon}\Big(\mathbb{E}[\hat{L}_\|(\hat{f}_\|)] + (4F + \sigma\sqrt{8\log 2n})\delta \\
&\quad + \Big(\frac{8\sigma^2}{n\epsilon} + \frac{8F\sigma\sqrt{2\log 2n}}{3n}\Big)(\log\mathcal{N}(\mathcal{F}_\|,\delta) + 1) - \frac{n-N}{n}\sigma^2\Big) + 4F\delta \\
&\le \frac{1}{1-\epsilon}\Big(\mathbb{E}[\hat{L}_\|(\check{f}_\|)] + (4F + \sigma\sqrt{8\log 2n})\delta \\
&\quad + \Big(\frac{8\sigma^2}{n\epsilon} + \frac{8F\sigma\sqrt{2\log 2n}}{3n}\Big)(\log\mathcal{N}(\mathcal{F}_\|,\delta) + 1) - \frac{n-N}{n}\sigma^2\Big) + 4F\delta \\
&\le \frac{1+\epsilon}{1-\epsilon}\mathrm{MSE}_\|(\check{f}_\|) + \frac{1}{n}\Big(\frac{8\sigma^2}{\epsilon} + \frac{8F\sigma\sqrt{2\log 2n}}{3}\Big)\Big(\frac{\log\mathcal{N}(\mathcal{F}_\|,\delta) + 2}{1-\epsilon}\Big) \\
&\quad + \Big(4F + \frac{4F + \sigma\sqrt{8\log 2n}}{1-\epsilon}\Big)\delta,
\end{aligned}$$

where the first line comes from (24), and second comes from (26), the thid line is because $\hat{f}_\| = \arg\min_{f \in \mathcal{F}_\|} \hat{L}_\|(f)$, and the last line comes from (25). We also use that fact that $\hat{L}_\|(\hat{f}) \le \hat{L}_\|(f), \forall f$. Noticing that $\mathrm{MSE}(\hat{f}) = \mathrm{MSE}_\|(\hat{f}) + \mathrm{MSE}_\perp(\hat{f})$, combining this with (23) finishes the proof. $\qquad\square$

## G  Detailed experimental setup

### G.1  Target Functions

The doppler function used in Figure 2(d)-(f) is

$$f(x) = \sin(4/(x + 0.01)) + 1.5.$$

The "vary" function used in Figure 2(g)-(i) is

$$\begin{aligned}
f(x) = {}& M_1(x/0.01) + M_1((x - 0.02)/0.02) + M_1((x - 0.06)/0.03) \\
& + M_1((x - 0.12)/0.04) + M_3((x - 0.2)/0.02) + M_3((x - 0.28)/0.04) \\
& + M_3((x - 0.44)/0.06) + M_3((x - 0.68)/0.08),
\end{aligned}$$

where $M_1, M_3$ are first and third order Cardinal B-spline bases functions respectively. We uniformly take 256 samples from 0 to 1 in the piecewise cubic function experiment, and uniformly 1000 samples from 0 to 1 in the doppler function and "vary" function experiment. We add zero mean independent (white) Gaussian noise to the observations. The standard derivation of noise is 0.4 in the doppler function experiment and 0.1 in the "vary" function experiment.

### G.2  Training/Fitting Method

In the piecewise polynomial function ("vary") experiment, the depth of the PNN $L = 10$, the width of each subnetwork $w = 10$, and the model contains $M = 500$ subnetworks. The depth of NN is also 10, and the width is 200 such that the NN and PNN have almost the same number of parameters. In the doppler function experiment, the depth of the PNN $L = 12$, the width of each subnetwork $w = 10$, and the model contains $M = 2000$ subnetworks, because this problem requires a more complex model to fit. The depth of NN is 12, and the width is 400. We used Adam optimizer with learning rate of $10^{-3}$. We first train the neural network layer by layer without weight decay. Specifically, we start with a two-layer neural network with the same number of subnetworks and the same width in each subnetwork, then train a three layer neural network by initializing the first layer using the trained two layer one, until the desired depth is reached. After that, we turn the weight decay parameter and train it until convergence. In both trend filtering and smoothing spline experiment, the order is 3, and in wavelet denoising experiment, we use sym4 wavelet with soft thresholding. We implement the trend filtering problem according to Tibshirani [40] using CVXPY, and use MOSEK to solve the convex optimization problem. We directly call R function $smooth.spline$ to solve smoothing spline.

### G.3  Post Processing

The degree of freedom of smoothing spline is returned by the solver in R, which is rounded to the nearest integer when plotting. To estimate the degree of freedom of trend filtering, for each choice of $\lambda$, we repeated the experiment for 10 times and compute the average number of nonzero knots as estimated degree of freedom. For neural networks, we use the definition [41]:

$$2\sigma^2 \text{df} = \mathbb{E}\|\boldsymbol{y}' - \hat{\boldsymbol{y}}\|_2^2 - \mathbb{E}\|\boldsymbol{y} - \hat{\boldsymbol{y}}\|_2^2 \tag{27}$$

where df denotes the degree of freedom, $\sigma^2$ is the variance of the noise, $\boldsymbol{y}$ are the labels, $\hat{\boldsymbol{y}}$ are the predictions and $\boldsymbol{y}'$ are independent copy of $y$. We find that estimating (27) directly by sampling leads to large error when the degree of freedom is small. Instead, we compute

$$2\sigma^2 \hat{\text{df}} = \hat{\mathbb{E}}\|\boldsymbol{y}_0 - \hat{\boldsymbol{y}}\|_2^2 - \hat{\mathbb{E}}\|\boldsymbol{y} - \hat{\boldsymbol{y}}\|_2^2 + \hat{\mathbb{E}}\|\boldsymbol{y} - \bar{y}_0\|_2^2 - \|\boldsymbol{y}_0 - \bar{y}_0\|_2^2 \tag{28}$$

where $\hat{\text{df}}$ is the estimated degree of freedom, $\mathbb{E}$ denotes the empirical average (sample mean), $\boldsymbol{y}_0$ is the target function and $\bar{y}_0$ is the mean of the target function in its domain.

**Proposition 17.** *The expectation of (28) over the dataset $\mathcal{D}$ equals (27).*

*Proof.*

$$2\sigma^2 \hat{\mathrm{df}} = \mathbb{E}_{\mathcal{D}}[\hat{\mathbb{E}}\|\boldsymbol{y}_0 - \hat{\boldsymbol{y}}\|_2^2 - \hat{\mathbb{E}}\|\boldsymbol{y} - \hat{\boldsymbol{y}}\|_2^2 + \hat{\mathbb{E}}\|\boldsymbol{y} - \bar{y}_0\|_2^2 - \|\boldsymbol{y}_0 - \bar{y}_0\|_2^2]$$

$$= \mathbb{E}\|\boldsymbol{y}_0 - \hat{\boldsymbol{y}}\|_2^2 - \mathbb{E}\|\boldsymbol{y} - \hat{\boldsymbol{y}}\|_2^2 + \mathbb{E}_{\mathcal{D}}[\hat{\mathbb{E}}[(\boldsymbol{y} - \boldsymbol{y}_0)(\boldsymbol{y} + \boldsymbol{y}_0 - 2\bar{y}_0)]]$$

$$= \mathbb{E}\|\boldsymbol{y}_0 - \hat{\boldsymbol{y}}\|_2^2 - \mathbb{E}\|\boldsymbol{y} - \hat{\boldsymbol{y}}\|_2^2 + \mathbb{E}\Big[\sum_{i=1}^{n} \epsilon_i(2y_i + \epsilon_i - 2\bar{y}_0)\Big]$$

$$= \mathbb{E}\|\boldsymbol{y}_0 - \hat{\boldsymbol{y}}\|_2^2 - \mathbb{E}\|\boldsymbol{y} - \hat{\boldsymbol{y}}\|_2^2 + n\sigma^2$$

$$= \mathbb{E}\|\boldsymbol{y}' - \hat{\boldsymbol{y}}\|_2^2 - \mathbb{E}\|\boldsymbol{y} - \hat{\boldsymbol{y}}\|_2^2$$

where $\mathcal{D}$ denotes the dataset. In the third line, we make use of the fact that $\mathbb{E}[\epsilon_i] = 0, \mathbb{E}[\epsilon_i^2] = \sigma^2$, and in the last line, we make use of $\mathbb{E}[\epsilon_i'] = 0, \mathbb{E}[\epsilon_i'^2] = \sigma^2$, and $\epsilon_i'$ are independent of $y_i$ and $y_{0,i}$ □

One can easily check that a "zero predictor" (a predictor that always predict $\bar{y}_0$, and it always predicts 0 if the target function has zero mean) always has an estimated degree of freedom of 0.

In Figure 2(h)(i), we take the minimum MSE over different choices of $\lambda$, and plot the average over 10 runs. Due to optimization issue, sometimes the neural networks are stuck at bad local minima and the empirical loss is larger than the global minimum by orders of magnitude. To deal with this problem, in Figure 2(h)(i), we manually detect these results by removing the experiments where the MSE is larger than 1.5 times the average MSE under the same setting, and remove them before computing the average.

### G.4 More experimental results

#### G.4.1 Regularization weight vs degree-of-freedom

As we explained in the previous section, the degree of freedom is the exact information-theoretic measure of the generalization gap. A Larger degree-of-freedom implies more overfitting.

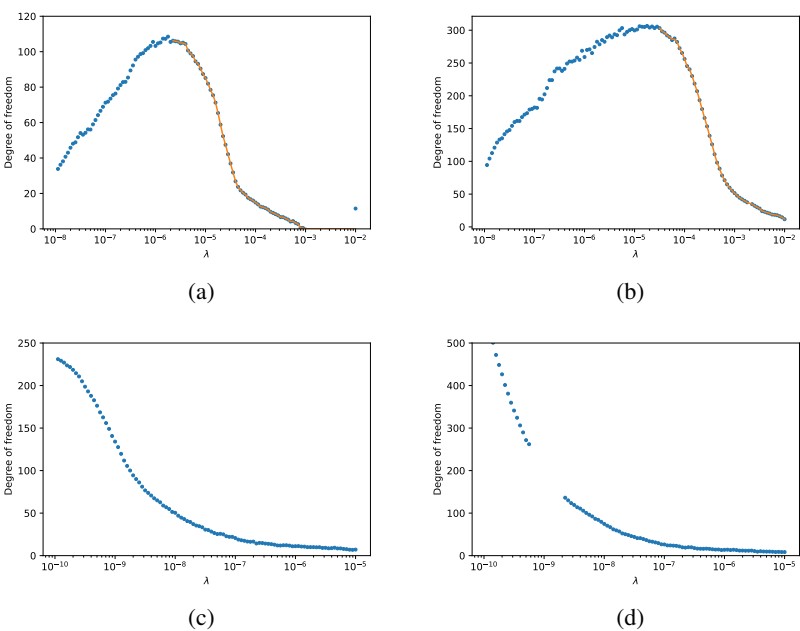

Figure 3: The relationship between degree of freedom and the scaling factor of the regularizer $\lambda$. The solid line shows the result after denoising. (a)(b)in a NN. (c)(d) In trend filtering. (a)(c): the piecewise cubic function. (b)(d) the doppler function.

In figure Figure 3, we show the relationship between the estimated degree of freedom and the scaling factor of the regularizer $\lambda$ in a parallel neural network and in trend filtering. As is shown in the

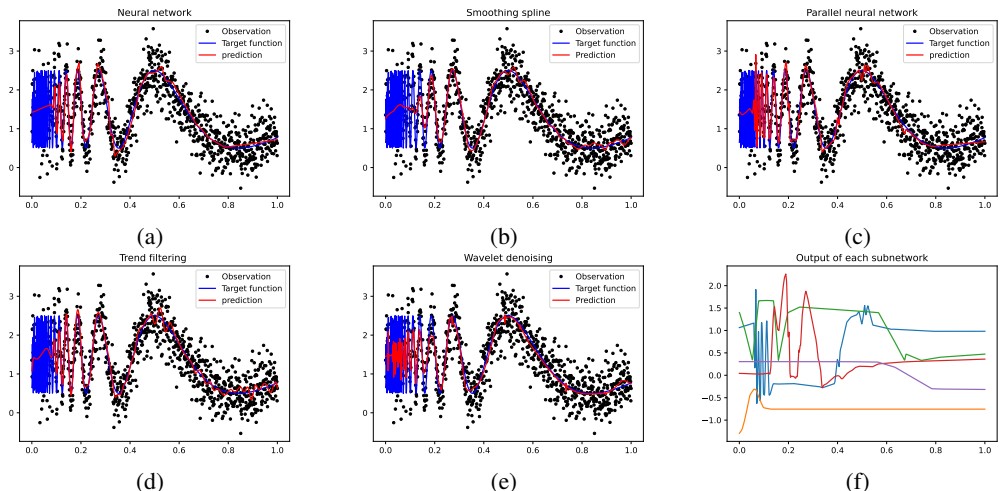

Figure 4: More experiments results of Doppler function.

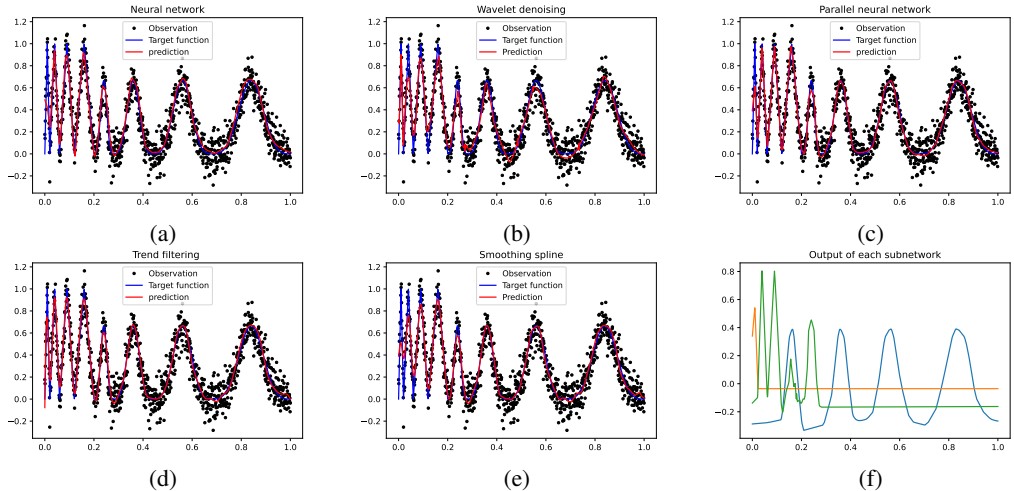

Figure 5: More experiments results of the "vary" function.

figure, generally speaking as $\lambda$ decreases towards $0$, the degree of freedom should increase too. However, for parallel neural networks, if $\lambda$ is very close to $0$, the estimated degree of freedom will not increase although the degree of freedom is much smaller than the number of parameters — actually even smaller than the number of subnetworks. Instead, it actually decreases a little. This effect has not been observed in other nonparametric regression methods, e.g. trend filtering, which overfits every noisy datapoint perfectly when $\lambda \rightarrow 0$. But for the neural networks, even if we do not regularize at all, the among of overfitting is still relatively mild $30/256$ vs $80/1000$. In our experiments using neural networks, when $\lambda$ is small, we denoise the estimated degree of freedom using isotonic regression.

We do not know the exact reason of this curious observation. Our hypothesis is that it might be related to issues with optimization, i.e., the optimizer ends up at a local minimum that generalizes better than a global minimum; or it could be connected to the "double descent" behavior of DNN [26] under over-parameterization.

### G.4.2 Detailed numerical results

In order to allow the readers to view our result in detail, we plot the numerical experiment results of each method separately in Figure 4 and Figure 5.

### G.4.3 Practical equivalence between the weight-decayed two-layer NN and L1-Trend Filtering

In this section we investigate the equivalence of two-layer NN and the locally adaptive regression splines from Section B. In the special case when $m = 1$ the special regularization reduces to weight decay and the non-standard truncated power activation becomes ReLU. We compare L1 trend filtering [22] (shown to be equivalent to locally adaptive regression splines by Tibshirani [40]) and an overparameterized version of the neural network for all regularization parameter $\lambda > 0$, i.e., a regularization path. The results are shown in Figure 6. It is clear that as the weight decay increases, it induces sparsity in the number of knots it selects similarly to L1-Trend Filtering, and the regularization path matches up nearly perfectly even though NNs are also learning knots locations.

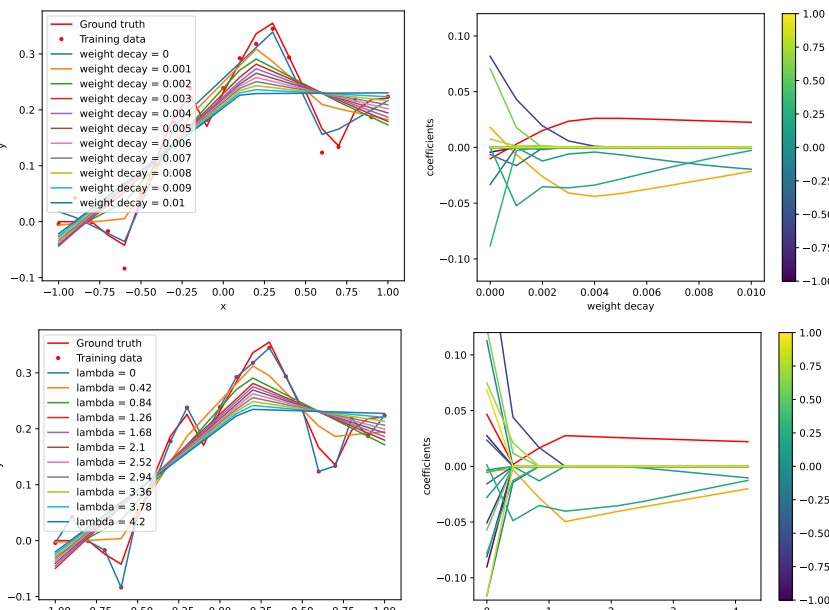

Figure 6: Comparison of the **weight decayed ReLU neural networks (Top row)** and **L1 Trend Filtering (Bottom row)** with different regularization parameters. The left column shows the fitted functions and the right column shows the *regularization path* (in the flavor of [17]) of the coefficients of the truncated power basis at individual data points (the free-knots learned by NN are snapped to the nearest input $x$ to be comparable).

