# OpenReview forum: "Deep Learning meets Nonparametric Regression: Are Weight-Decayed DNNs Locally Adaptive?"
_NeurIPS.cc/2022/Conference — NeurIPS 2022 Submitted_

### Official Review · Reviewer_D3YQ · 2022-07-06

**Rating:** 5
**Confidence:** 4
**Soundness:** 3 good
**Presentation:** 3 good
**Contribution:** 3 good

**Summary:**

1. This paper considers the mean regression using the parallel ReLU activated neural networks.

2. This papers observes that the training is equivalent to $\ell_p$ sparse regression with a learned basis.

3. The paper establishes the error bound for MSE($\hat f$), which almost achieves the minimax rate of a Besov class when the depth $L$ is large.

**Questions:**

1. The method highly depends on parallel networks. This is different with general usage of DNNs. So the conclusion of this paper cannot address the success of deep learning in a general way.

2. It is not very intuitive that the $\ell_p$ sparse regression with a learned basis can adapt to unknown smoothness. Can the authors use the results from Parhi and Nowak (2021) to address further properties of each learned basis?

3. The paper claims that their regression problem has a fixed design, so data points are not iid. Can you clarify the meaning of this? Do iid data points make the problem easier?

4. The numerical experiments are over-simplified and only consider the 1d regression. For example, for an easy MNIST dataset, how large $M$ should we use?


**Limitations:**

There is no potential negative societal impact of their work.

**Strengths And Weaknesses:**

Originality: The connection between deep learning and nonparametric statistics is an important question. The recent papers by Parhi and Nowak (2021) and Suzuki (2018) are inspiring. This paper combines a couple of previous techniques and provides some promising results.

Quality: This is a theoretical paper and it is technical sound. The claims are well supported.

Clarity: The submission is clearly written and well organized.

Significance: The results are important. However, the experiment section is not very convincible. The practical usage of proposed method is questionable.

---

> ### Author Response · Authors · 2022-08-02
> **Response to reviewer D3YQ**
>
> Thanks for your effort in reviewing the paper and thoughtful comments. Our response to the comments are below:
>
> 1. The numerical results seem to suggest vanilla DNN with weight-decay is not locally adaptive, i.e., does not achieve optimal error rates for estimating BV and Besov functions. Comparing to the sparsity constraint used by Suzuki, we find parallel NN more practical for optimization and naturally provides a compressed model at deployment time (only a small number of subnetworks are active).
>
> 2. The reason why Parallel DNN can adapt to unknown smoothness (while classical methods and shallow NNs cannot) is representation learning. For each order of smoothness, the PNN could approximate the most appropriate set of basis functions to span functions in that class. Specifically, in the approximation-theory part of the proof, we showed that each subnetwork can uniformly approximate the truncated power function $(x-x_i)_+^m$ for any knot $x_i$ and smoothness order $m$. A constant number of these truncated power functions can approximate the B-spline wavelets of any constant order, thereby approximating all Besov class functions.
>
> We did not use techniques for Parhi and Nowak for proving this fact. Instead, this builds upon Dung (2011) and Suzuki (2018). We'd appreciate if you can provide more specific pointers to Parhi and Nowak so we can look into it further for connections.
>
> Regarding the exact form of the learned basis, it is a bit difficult to determine. Our Figure 2(c)(f) gave a few empirical examples of them.  However, our approximation-theory results provide certificates such that no matter what basis functions PNN manage to learn, they are no worse than the B-spline wavelets for representing the particular input function. This partially explains why in practice PNN works better than wavelet denoising.
>
> 3. By fixed design, we mean that the covariates $\{x_i\}$ are fixed, and the expectation is taken only over $\{y_i\}$ (or $\{\epsilon_i\}$). This setting is typical in non-parametric regression literature and we choose to focus on it to be comparable. Results in the fixed design setting do not directly imply or be implied by the results in the iid setting, though many low-level techniques in the paper can be used for the iid case too. Handling the fixed design version of the problem required us to work out the corresponding statistical learning theory tools, e.g., self-bounding arguments for this setting, which could be of independent interest.
>
> 4. The main theme of the paper is to study DNNs in nonparametric regression problems, which is often about estimating low-dimensional functions in very broad function classes. Our experiments are thus designed to illustrate this particular aspect. The focus is to validate our theoretical findings, e.g., sparsity and local adaptivity, and to compare to the SOTA nonparametric regression methods such as trend filtering in their own turfs. We believe our experiments, while simple, have achieved our goals. We think Parallel NN architecture should work well on high-dimensional data (such as MNIST) too. It however requires very different set of assumptions on the function classes to overcome the curse of dimensionality, thus is beyond the scope of the current paper.
>
> We do however wanted to emphasize that while the statistical analysis currently focuses on non-parametric regression tasks, the other part of our result --- the equivalence between Weight-Decay Parallel DNN and a sparse regression problem with representation learning --- is directly applicable to the more general settings (e.g., computer vision / NLP tasks, cross entropy losses).
>
> Thank you again for the thoughtful review of our work. We hope our responses satisfactorily addressed your questions. If so, we would really appreciate if you could consider raising the score.

---

> > ### Author Response · Authors · 2022-08-07
> > **Do you have further questions / suggestions?**
> >
> > Thanks again for the kind review!  As our chance to post any messages is about to expire, may I know if there is anything you'd like us to clarify in our response to the questions?

---

### Official Review · Reviewer_RAZ6 · 2022-07-09

**Rating:** 4
**Confidence:** 4
**Soundness:** 2 fair
**Presentation:** 2 fair
**Contribution:** 2 fair

**Summary:**

This paper studies weight decay regularized neural network training problems through the lens of nonparametric regression. The authors particularly consider a parallel architecture that is a weighted combination of multiple standard networks and show that weight decay regularization promotes Lp sparsity for the coefficients of the learned dictionary. Thus, they aim to shed light on why depth matters and the distinction between the expressive power of neural networks and kernel methods.

**Questions:**

* I don’t see any theoretical reasons for studying a parallel architecture rather than a standard deep neural network. It seems that the authors already allocated a section to prove common usage of this architecture in various theoretical and empirical works. But, I still think they need to explicitly describe the reasons for choosing this specific architecture and explain why analyzing standard networks are challenging if that is the case.
* Could the authors comment on the impact of regularizing the bias? Even though this is the common practice for numerical experiments, it might make the problem trivial when one aims to globally optimize the objective function rather than using certain optimizers, such as SGD, with benign implicit regularization effect. In other words, if the network is deep enough one can choose arbitrarily small layer weights satisfying the constraints in Proposition 1 and then optimize over biases to exactly fit the training data.
* It seems that this kind of a sparsity-inducing properties of weight decay regularized NNs have already been studied in [1,2]. I think the authors should comment on these studies and clarify their contributions.
* Regarding the numerical experiments, I am wondering if the comparison between parallel NN and standard NN is fair. For instance, what is the number of parameters for each architecture? What is the depth of the networks? Which optimizer did you use? I think the authors should provide more details in the numerical experiments section.
* Is it possible to extend this analysis to different activations, e.g., smooth activations such as tanh, sigmoid or other nonsmooth activations such as leaky ReLU, piecewise linear activations?
* Do these conclusions apply to other NN architectures such as CNN? It seems that [3] has already proven that two-layer weight decay regularized CNNs also induce a sparsity across filters. Does this work have similar implications possibly for deeper CNNs?

[1] "Neural networks are convex regularizers: Exact polynomial-time convex optimization formulations for two-layer networks.", ICML 2020

[2] "Global optimality beyond two layers: Training deep relu networks via convex programs." ICML 2021

[3] "Implicit convex regularizers of cnn architectures: Convex optimization of two-and three-layer networks in polynomial time." ICLR 2020



**Limitations:**

This is mostly a theory paper on neural networks therefore doesn't have any negative societal impacts.

**Strengths And Weaknesses:**

The paper is mostly well written and the results seem theoretically sound, however, there are some point requiring further clarifications as detailed below:

---

> ### Author Response · Authors · 2022-08-02
> **Respond to reviewer PAZ6**
>
> Thanks for your effort in reviewing the paper and thoughtful comments. Our response to the comments are below:
> 1. Parallel NN is just a special vanilla NN with a pre-defined block-diagonal sparsity pattern. The reason we study parallel neural networks rather than vanilla neural networks is that parallel neural networks trained with weight decay is equivalent to an $\ell_p$ sparse model, which is the key that it can achieve close to minimax rate in Besov class. Although we did not prove explicitly, the numerical results suggest that vanilla neural networks do not induce sparsity and are not (close to) minimax optimal. While other works (Suzuki, 2018) on error rate of neural networks do not require a parallel neural network structure, they instead require an L0-sparsity constraint on the Neural Network weights. We think it is harder to train and tune a NN with $\ell_0$ sparsity constraint, than training a parallel neural network with standard weight-decay.
>
> While parallel NNs are less commonly used right now, they have many interesting properties that make them more appealing than the standard NN. One implication of our results is that PNNs naturally learn a deploy-efficient model (since most subnetworks will predict constants thanks to the implicit sparse regularization of the weight decay and can be removed; see Fig 2 (c)(f) for the only subnetworks that are active in our experiments).
>
> 2. We believe the question is about why we do *not* regularize the biases (the original question has a typo)? If we regularize them instead, neural networks will be biased in predicting the constant term $f(x)=c$ in the target function.
> From a theoretical prospective, our theory suggests that leaving out the biases from weight-decay is actually important for achieving near-optimal rates for non-parametric regression, and regularizing the bias will affect our ability to compare neural networks with other non-parametric regression methods, as most of those methods have a similar null space, (function space that is not regularized).
> From the technical perspective, handling these unregularized biases is one of our key technical contributions because they are not constrained and do not have a bounded covering number.
> From an empirical prospective, this is not an issue as data are often normalized before feeding into a neural network.
>
> 3. We thank the reviewer for suggesting the references. The two papers focus on two-layer and three-layer neural networks, respectively, while our work focus on arbitrarily deep (parallel) neural network. Besides, our paper provides the upper bound of error rate with respect to the number of training samples, which is not given in these works. We have added the reference in the revised version of the paper.
>
> 4. We tried our best to make sure the comparison is fair by choosing the architecture such that they have the same depth $L$ and approximately the same number of parameters. Details of the experiments are listed in the appendix G.2.
>
> 5. It is a good question. We are not sure about the answer, as these activation functions are not $1$-homogeneous.
> For leaky-ReLU and piecewise linear activation, it is clear that they are spanning the same function class as ReLU, but the implicit regularization in the function space induced by weight decay will be quite different. It is a good direction of future work to work these out explicitly.
>
> 6. It may apply but we are not sure. We have not looked at CNNs but it is definitely an interesting future direction
>
> We hope our responses clarified our novelty and satisfactorily addressed your concerns. We would appreciate if the reviewer could consider raising the score.

---

> > ### Author Response · Authors · 2022-08-07
> > **Further questions on our response?**
> >
> > Thanks again for asking insightful questions.  We tried our best to clarify above. May I know if our responses make sense?
> >
> > We know that PNN vs NN might be a subjective discussion, but do you find our added discussion on the differences from the related work [1][2][3] you suggested fair and reasonable?  Please let us know if we missed anything obvious...
> >
> > Thank you very much!

---

> > > ### Comment · Reviewer_RAZ6 · 2022-08-07
> > > **Re:**
> > >
> > > I want to thank the authors for their detailed response but I still have some questions:
> > >
> > > 1. In the introduction section, you claim that these parallel architectures are important since they are also proven to be work well in practice. However, the referred neural network (NN) architectures, e.g., SqueezeNet, ResNext and Inception, use a different NN for each subnetwork. Do you have any implications for a case where subnetworks are not identical? In addition, these architectures perform well since they utilize the benign impact of architectural choices, such as convolutional structures, however, the analysis in this paper only applies to fully connected networks. Although I appreciate the theory for fully connected networks, I am not sure if this will lead to any useful practical implications in future works. Do you have any comments on this?
> > >
> > > 2. What do you mean by "neural networks will be biased in predicting the constant term in the target function"? People usually put weight decay regularization during training and that significantly boosts the generalization performance of the function learned by an NN. What is the difference in this case? Could you please further clarify this statement?
> > >
> > > 3. I checked the number of parameters for each architecture in the numerical experiments section. But it seems that you used significantly more parameters for the parallel NNs than standard NNs. Could you please provide precise number of parameters for each architecture in your experiments and explain why you think that this setup is fair?

---

> > > > ### Author Response · Authors · 2022-08-07
> > > > **Respond to reviewer PAZ6**
> > > >
> > > > Thanks for the further questions.
> > > >
> > > > 1.  Our results do not directly apply to other subnetwork architectures as stated. But some of the techniques may work, e.g., a Conv Layer is also a linear matrix multiplication, thus with ReLU, we can use the same technique for deriving the equivalence to L_p sparsity. Other more esoteric layers, e.g., BatchNorm, do not seem to work.  Overall, as a primarily theoretical work, the direct implications of our work to practical NN architecture design are limited. Though we do hope that the system benefits of PNN will be picked up and developed further.
> > > >
> > > > 2. We did not mean that one shouldn’t use weight decay. Quantifying the benefits of weight decay is a main contribution of the work. We thought you were asking about weight decay on the bias part of the parameters in each layer or not.
> > > > We learned from the Elements of Statistical Learning book that L2 regularization should be applied to the weights but not the bias parameter in the linear models. We believe this is also a standard choice in DNN too, e.g., Page 226 of the Goodfellow et al book on Deep Learning (https://www.deeplearningbook.org/contents/regularization.html) (we quote below):
> > > > >“we note that for neural networks, we typically choose to use a parameter norm penalty $\Omega$ that penalizes only the weights of the aﬃne transformation at each layer **and leaves the biases unregularized.** ”.
> > > >
> > > > Modern neural networks implementations seem to adopt this convention too, eg. bert  https://github.com/huggingface/transformers/blob/main/src/transformers/trainer.py#L1041 .
> > > >
> > > > What we found is that this choice is actually important when applying DNN to nonparametric regression. As an example, if biases are left unregularized, the model will fit f(x) and f(x) +  1e20  identically, whereas if we regularize the bias then the model will output a very poor fit for the latter.
> > > >
> > > > 3. The number of parameters in a parallel NN can be roughly estimated as $L\times M \times w^2$ where $L$ is the depth of the NN, $M$ is the number of subnetworks, $w$ is the width of NN. The number of parameters in a  NN can be roughly estimated as $L \times w^2$. You can check in both experiments, they are approximately the same.
> > > > On the other hand, we recognize that there is no perfect way of comparing the “complexity” of two different function classes, especially when regularized. Remind that the parallel NN is sparse after training so it contains far fewer number of nonzero parameters than before training. The number of parameters is often a bad surrogate.  For this reason, we aligned the “effective degree of freedom” in Figure 2 (b) (e) so they are comparable (note that to achieve the same dof, PNN and NN might need to use very different weight decay parameter).
> > > >
> > > > As a reference, we provide the exact number of parameters (including zero parameters) in the experiments below:
> > > >
> > > > In dopler experiment, the PNN we use contains:
> > > > > Weight in the first layer: $1\times 10\times 2000=20000$
> > > > > Bias in the first layer: $1\times 10\times 2000=20000$
> > > > > Weight in the last layer: $10\times 1\times 2000=20000$
> > > > > Bias in the last layer: $1$
> > > > > Weight in the rest layers: $10\times 10\times 2000=200000$   (there are 10 such layers)
> > > > > Bias in the rest layers: $10\times 2000=20000$ (there are 10 such layers)
> > > > > Total number of parameters:  $2260001$
> > > >
> > > > The NN we use contains
> > > >  > Weight in the first layer: $1\times 400=400$
> > > > > Bias in the first layer: $400$
> > > > > Weight in the last layer: $400\times 1=400$
> > > > > Bias in the last layer: $1$
> > > > > Weight in the rest layers: $400\times 400=160000$   (there are 10 such layers)
> > > > > Bias in the rest layers: $400$ (there are 10 such layers)
> > > > > Total number of parameters:  $1605201$
> > > >
> > > > In "vary" experiment, the PNN we use contains:
> > > > > Weight in the first layer: $1\times 10\times 500=5000$
> > > > > Bias in the first layer: $1\times 10\times 500=5000$
> > > > > Weight in the last layer: $10\times 1\times 500=5000$
> > > > > Bias in the last layer: $1$
> > > > > Weight in the rest layers: $10\times 10\times 500=50000$   (there are 8 such layers)
> > > > > Bias in the rest layers: $10\times 500=5000$ (there are 8 such layers)
> > > > > Total number of parameters:  $455001$
> > > >
> > > > The NN we use contains
> > > >  > Weight in the first layer: $1\times 200=200$
> > > > > Bias in the first layer: $200$
> > > > > Weight in the last layer: $200\times 1=200$
> > > > > Bias in the last layer: $1$
> > > > > Weight in the rest layers: $200\times 200=40000$   (there are 8 such layers)
> > > > > Bias in the rest layers: $200$ (there are 8 such layers)
> > > > > Total number of parameters:  $322201$
> > > > You can see that the number of parameters in parallel NN and NN are approximately the same in each experiment.

---

> > > > > ### Author Response · Authors · 2022-08-10
> > > > > **Clarification about the experiment, and experiment plan**
> > > > >
> > > > > We realized that the reviewer might be referring to the ~30% smaller number of parameters in NN comparing to PNN.  That is not intentional and we were only trying to roughly match them in a ballpark. We will re-do the NN experiments by increasing the width to 240 and 470 respectively, such that the difference is <2%.  We will add the figure if the AC / Openreview allows us to.  If not, we will add them to the final version.
> > > > > Based on our theoretical understanding of these models, we do not think it will really change anything because both NN and PNN contain substantially more parameters than what is necessary to fit those simple underlying functions.  The current experiments should be convincingly supporting our theoretical results on PNN vs classical nonparameteric regression methods.

---

### Official Review · Reviewer_yJWs · 2022-07-18

**Rating:** 6
**Confidence:** 1
**Soundness:** 3 good
**Presentation:** 3 good
**Contribution:** 3 good

**Summary:**

This submission first shows the equivalence between standard weight decay and sparse penalty in the parallel neural networks. The authors then show that the estimation error rate is close to the minimax one. Finally, simulation results are provided to compare the approximation from different estimators.

**Questions:**

See above

**Limitations:**

Yes

**Strengths And Weaknesses:**

This submission first shows the equivalence between standard weight decay and sparse penalty in the parallel neural networks. The authors then show that the estimation error rate is close to the minimax one. Finally, simulation results are provided to compare the approximation from different estimators.

*Strengths And Weaknesses
First I need to say that I am not an expert on the topics of this submission, and hope other reviewers with relevant background can comment more on the significance and novelty for the theoretical results.

Strength:
One interesting observation is that the authors proved that deeper models achieve smaller MSE error, and hence is closer to the minimax rate. Although this is only for the special parallel ReLU DNNs, it is still helpful to gain a better understanding on the empirical better performance of deeper models.

Weakness:
A few parts of this paper are a bit difficult to follow, and some clarification would be helpful, which are listed as follows.
1. I am not sure how sparsity is shown in the equation in Figure 1 (b). If L \neq 2 and hence it's not an L-1 norm, how can we still obtain the sparse regression?

2. The purpose of the results in Figure 2 is a bit unclear. For example, the authors claimed parallel NN achieves smaller error with deeper networks. Could you provide any results to support this claim? Furthermore, unlike wavelet denoising, why does NN tend to achieve higher MSE with higher DOF in the end of curves?

---

> ### Author Response · Authors · 2022-08-02
> **Respond to reviewer yJWs**
>
> Thanks for your effort in reviewing the paper and thoughtful comments. Our response to the comments are below:
> 1. When $L>0$, $L-p$ (quasi)-norm is used where $0<p<1$. This norm is known to induce sparsity as well. See also https://ieeexplore.ieee.org/document/9048923 .
> To quickly convince yourself, take $p\rightarrow 0$ and you will see that it becomes "the number of non-zero elements" --- sparsity by definition.
>
> 2. The purpose of Figure 2 is to demonstrate that parallel neural networks can achieve practical performance comparable to the SOTA locally adaptive nonparameteric regression methods for estimating functions in the BV class. This is quite surprising because we are also learning the representation, rather than hard-coding the basis functions as in classical non-parametric regression methods.
> Plotting MSE vs effective DoF is the standard way of aligning the complexity of the learned function due to changing regularization weights used in different methods, see Figure 6 of Tibshirani, "Adaptive piecewise polynomial estimation via trend filtering".
> By definition, DoF $\propto$ Expected Generalization Gap. So for any method, larger DoF would mean we are fitting a more complex function to the data which gives smaller (square) bias and larger variance. The MSE is the sum of the two, thus the U-shaped curve.
> Generally speaking, large regularization weight gives smaller DoF. For generalized lasso (including Trend Filtering), DoF is the expected number of non-zero coefficients selected. For neural networks (standard and parallel versions), both the learned representation and the fitted coefficients contribute to the DoF and they required us to use a novel technique to accurately estimate it.
> It is a good question why (non-parallel) NN does not work as well and does not see the same reduction in the biases as we increase DoF. However, it is encouraging that PNN works competitively as our theory predicts, despite needing to fit significantly more parameters than the classical methods.

---

### Official Review · Reviewer_2BVk · 2022-07-18

**Rating:** 6
**Confidence:** 3
**Soundness:** 2 fair
**Presentation:** 2 fair
**Contribution:** 2 fair

**Summary:**

The paper studies how weight-decayed parallel ReLU networks are able to estimate functions with heterogeneous smoothness in the Bounded Variance and Besov classes. In particular, the authors show that the weight decay in the L-th layer parallel network is equivalent to a sparse $\ell_{L/2}$ constraint and use this property with an upper bound of the covering number to bound the estimation error. The authors then provide an approximation error bound by using the B-spline basis functions as a proxy. They have shown that for a deep network, the error rate is nearly optimal.

**Questions:**

1. In the extended form of theorem 1 in the appendix, the dominant term of the MSE also contains a large dependency over the depth $L$. When $L$ is large, I would assume this term to be linear in $L$. As a result, when $L$ is large, the constant term would decrease exponentially with $L$, but the first and dominant term grows linearly in $L$. I feel this cannot support the statement that deeper parallel networks achieve lower or even optimal error rates. Is there somewhere I was misleading here?

2. Related to the comments above, how large are $\lambda$ and $P'$?

**Limitations:**

The authors have adequately addressed the limitations and potential negative societal impact of their work.

**Strengths And Weaknesses:**

The paper provides the first optimal rate approximation results of a weight decay parallel ReLU network over Besov functions. The observation that the weight decay in parallel networks is equivalent to a sparse promoting constraint is interesting. It enables the control of the complexity of the model, which is characterized by the covering number in the paper. This also allows the number of subnetworks $M$ to be large without overfitting. The paper also shows that each subnetwork can adapt to functions with different local smoothness properties (different B-spline basis functions) and thus have the ability to be locally adaptive.

Weakness:

1. The main complaint is that the statements of the theorems and propositions are not provided in a clear and rigorous way. Theorem 1 is confusing in the sense that it does not specify the assumption for the target function $f_0$. There seem to be two sets of function classes considered in section 2.2, the Besov spaces $B_{p, \infty}^{\alpha}$ and bounded variation space $BV(m)$. From the statement, I would assume the target function belongs to $B_{p, \infty}^{\alpha}$, is that true? And how is $m$ defined in the problem? Is $m$ to be any integer with $m > \alpha$? $r > 0$ is assumed in the statement but $r$ does not appear anywhere else in the theorem. Also, I find the statement "with proper choice of the parameter of weight decay $\lambda$" to be vague. It would be helpful if how $\lambda$ depends on other parameters in the theorem can be specified. The authors provided an extension version of theorem 1 in appendix F, but it does not resolve the problems above. For example, the proof also uses "there exists a weight decay parameter \lambda'" in line 775, which causes confusion as I would like to know how large this $\lambda$ or $\lambda'$ would be.

There are similar unclear statements in other parts of the paper. In proposition 3, the authors do not specify how $P'$ depends on the other parameters. In fact, $P'$ would depend on $\lambda$, and may not be viewed as a constant. Actually, in proposition 6, one can see that $a$ can have a large $2/L$ norm that depends exponential over $m$. In theorem 4, the notation of $\log \mathcal N(\mathcal F, \delta)$ is confusing as $\mathcal F$ is used to denote the model defined in (5) apart from the bias in the last layer. In proposition 6, $M_{m, k, s}$ is used but its definition is only shown later in proposition 7.

All these unclarities in the theorem statement (especially the ones in theorem 1 and proposition 3) make it hard to either examine the theorem or evaluate its contributions appropriately.

2. The paper uses parallel networks instead of typical neural network structures. As authors have pointed out, weight decays in the parallel networks would result in sparse promoting networks, but vanilla neural networks don't seem to have the same behavior. Also, the paper talks about approximation ability instead of that of training, and as the results rely on the homogeneity of ReLU, it is unclear whether a trained weight decay parallel network would also find a sparse solution.

3. The experiments do not support the arguments nicely. In figure 2, the nonsmooth part (the left part in figure (a)(d)) are hard to see clearly and thus impossible to tell which methods perform better. Also, for a highly oscillating function, one would need a larger number of samples to be able to fit it correctly, which may be the reason why figure (b) and (c) looks noisy. It would be helpful in plotting the relationship between the MSE and the sample sizes instead of the degree of freedom. In figure (c)(f), it is hard to see the sparsity of the subnetworks well. From the appendix, there seem to be $M = 2000$ subnetworks. It would be helpful in providing the scale of each subnetwork in total instead of plotting a few of these subnetwork outputs.


Typos:
1. equation (4) missing a parenthesis on the left-hand side
2. the axis in figure 2 is not shown properly ("MSE" in (b)(d) is partially covered)

---

> ### Author Response · Authors · 2022-08-02
> **Response to reviewer  2BVk**
>
> Thank you for the detailed review! We will first clarify the theorem statement
> and comment on how $\lambda$ and $P'$ are connected to each other, then we will address
> your questions on the experiments.
>
> Respond to comments:
> 1. We apologize for the editting issue in Theorem 1 and others. We've corrected them in the revised version.
> In proposition 3, we did not argue that $P'$  is a constant. Actually it depends
> on not only $\lambda$  but also $\alpha, f_0$ and $\mathcal{D}_n$. The inverse map from $P'$ to $\lambda$ is a
> complex function that comes from Lagrange duality (Proposition 3). The key
> point is that for any set of $\alpha, f_0$ and $\mathcal{D}_n$, there is a one-to-one mapping between
> $\lambda$  and $P'$ . In practice, what this means is that tuning hyperparameter $\lambda$  is
> equivalent to tuning hyperparameter $P'$  in the re-parameterized form. Our
> subsequent analysis thus focused on the constrained form. In the proof of the
> main theorem, Line 781 (Line 811 in the revised version), we have provided a specific choice of $P'$  that yields the
> stated MSE bounds.
> 2. In this paper, we only focus on the argmin estimator but not the training
> procedure. The implicit bias caused by training procedure is of separate interest.
> On the other hand, as is demonstrated in figure 2 where we plotted all the
> subnetworks that output a nonconstant value, the number of such “active” subnetworks
> is much smaller than the total number of subnetworks. This indicates
> that a trained weight decay neural network can also find a sparse solution, but
> the basis function of the trained sparse solution is not necessarily the one we
> used in the proof.
> 3. In figure 2 (c)(f), we plot all the subnetworks that is not a constant for
> $x \in [0, 1]$ after training, so the sparsity is clear from the figure.
>
> Questions:
> 1. When $n$ is large (often millions or larger), and $L$ is relatively small (often less than 100),
> which is typical for neural networks, the first term decreases as $L$ increase.
> On the other hand, one can choose $L=O(\log n)$, in this case MSE then scales
> as $\tilde{O}(n^{-\frac{2\alpha}{2\alpha+d}(1 - o(1))})$, where the $o(1)$ is $O(1/\log n)$.
> See also proposition 2 in the revised version.
> 2. Please see our response about the first comment.

---

> > ### Author Response · Authors · 2022-08-07
> > **Did our responses address your concerns?**
> >
> > Thanks again for writing a detailed review!  May I know if our response above clarified the questions on (1) the dependence on L,  (2) Sparsity in Figure (c)(f); and (3) the choice of $\lambda$ (actually $P'$ in the dual form)?
> >
> > We are sorry for the ping over the weekend but we would like to to have a chance to respond in case you have further questions.

---

> > > ### Comment · Reviewer_2BVk · 2022-08-08
> > > **Response to the authors**
> > >
> > > I thank the authors for their detailed and helpful responses.
> > >
> > > The new version of theorem 1 is way much better than the earlier version. It is now in the form that one can understand what the theorem claims.
> > >
> > > My concern for figure 2(f) is addressed. It is just a bit surprising that the parallel network with 2000 subnetworks can have sparsity 5.
> > >
> > > For question 1, I agree with the authors that in many settings, sample size $n$ can be a few magnitudes larger than the depth $L$. But the assumption $L = O(\log n)$ is still not reasonable.
> > >
> > > For question 2, I thank the authors for helping to clarify the relationship between $P'$ and $\lambda$. I would encourage the authors to state this one-to-one mapping between $\lambda$ and $P'$ more explicitly in the paper. Currently, it is partially shown in proposition 3, but it goes from $\lambda$ to $P'$, while what is needed is the other direction.
> > >
> > > Based on the discussion and the new version of the paper, I am willing to increase my original rating.
> > >
> > > minor:
> > > 1. In corollary 2, there is a $n$ missing. $C \log n \le L \le 100 C \log n$.
> > > 2. In line 573, Langrange’s method should be Lagrange’s method?

---

> > > > ### Author Response · Authors · 2022-08-08
> > > > **Thanks!**
> > > >
> > > > >The new version of theorem 1 is way much better than the earlier version. It is now in the form that one can understand what the theorem claims.
> > > >
> > > > Thank you for your feedback that leads to the improvements of the paper.
> > > >
> > > > > My concern for figure 2(f) is addressed. It is just a bit surprising that the parallel network with 2000 subnetworks can have sparsity 5.
> > > >
> > > > We found it very surprising that theory checks out in the experiments so well too.  This in some sense demonstrates that PNN is really able to adaptively select the most appropriate complexity to fit the data.
> > > >
> > > > > For question 1, I agree with the authors that in many settings, sample size can be a few magnitudes larger than the depth $L$. But the assumption is still not reasonable.
> > > >
> > > > The $e^{-c L}$ term in the bound (that Const.) is due to the approximation error of very smooth functions using ReLU DNN, which is known to only give piecewise linear functions. So $L = O(\log n)$ is what makes this term smaller than the optimal rate.   We believe it is not improvable due to the non-differentiability of ReLU.  Notice that if we allow smoother activation functions, then even $L = 2$ works (see Section B in our appendix and the corresponding experiments in Section G.4.3.), though the NN there is more esoteric and less used in practice.
> > > >
> > > > > For question 2, I thank the authors for helping to clarify the relationship between  $P'$ and $\lambda$. I would encourage the authors to state this one-to-one mapping between $\lambda$ and $P'$ more explicitly in the paper. Currently, it is partially shown in proposition 3, but it goes from $\lambda$ to $P'$, while what is needed is the other direction.
> > > >
> > > > Thank you! We will add a remark to clarify Lagrange’s method and the one-to-one mapping.
> > > >
> > > > > Based on the discussion and the new version of the paper, I am willing to increase my original rating
> > > >
> > > > Thank you for championing the paper! This is greatly appreciated.

---

### Author Response · Authors · 2022-08-02
**Respond to all the reviewers, clarification and summary of revision.**

We thank the reviewers for their thoughtful comments and time spent on
our paper. We appreciate the reviewers for correctly highlighting interesting
aspects of the work such as:
* The equivalence of weighted decayed parallel DNNs to an $\ell_p$-sparse linear
regression with representation learning.
* A formal separation between kernel methods (e.g., NTK) and neural networks.
* The provably benefit of having deeper NNs, and a new explanation for
moderate overparameterization.
* Near-optimal error rates for estimating functions with heterogeneous smoothness.

We would like to also thank the reviewers for the questions and concerns,
which we have addressed in the revision and in the individual response below
each review.
In the revision, we resolved the clarity issues raised by Reviewer 2BVk.
The statement of Theorem 1 is now self-contained, and the descriptions of the
experiments are updated such that we explicitly state how they support the
theoretical findings. Regarding the dependence on $L$, we added a corollary that
sets $L = O(\log n)$, which gives a simplified MSE bound of
$\tilde{O}(n^{-\frac{2\alpha}{2\alpha +d}(1-o(1))} )$
for Besov class functions. This more clearly demonstrates how the results nearly
match the lower bounds.
We have also discussed the additional references Reviewer RAZ6 suggested
and confirmed that they do not cover any of our novel contributions.
Regarding the concerns on the significance and relevance of parallel NN
architecture raised by Reviewer 2BVk, RAZ6 and D3YQ, we would like to make
the following observations (in addition to what we have already covered in Line
52-56):
1. Parallel NN is can be viewed as a vanilla DNN with a block-diagonal
constraints on the weights. In comparison, Suzuki (2018) requires a general $\ell_0$
constraint, which is harder to implement.
2. Neither our work nor Suzuki (2018) answers the open problem of whether
vanilla dense DNNs with weight-decay can be locally adaptive. But our numerical
experiments seem to suggest a negative answer to the conjecture, which
would imply that parallel NNs can be better than vanilla NNs for the family of
problems we consider.
3. Parallel NN is an interesting neural architecture on its own right! Our
results show that it enjoys a highly desirable property of naturally learning a
compressed model where only a very small number of subnetworks will be active.
This makes it more memory efficient in deployment-time. This reminds us of
the “lottery ticket hypothesis” (see Fig 2 (c)(f) for the only subnetworks that
are active).

To summarize, parallel NN indeed differs from standard NN and we do not
imagine that parallel NN to be more popular than standard NN any time soon.
However, in light of its interesting properties above, maybe it is better for the
community to give new ideas like this a chance to grow?
We hope the reviewers find our responses clarifying and that all key concerns
are convincingly addressed. If so, we would appreciate if the reviewers would
consider raising the score. We are happy to respond to any further questions and
discussion in the meantime. Again many thanks for your time and attention

---

### Meta-Review · Area_Chair_Wgdj · 2022-08-27

**Recommendation:** Reject
**Confidence:** Certain

**Metareview:**

This paper considers the generalization error of neural networks when approximating a specific set of functions. The main contribution of this paper is showing a rate of neural network generalization error which is better than linear functions.


The major problems of this paper are

(1). Missing an important sequence of works on separating the power of neural networks from kernel methods, for example, "what can ResNet efficiently learn, going beyond kernels". The authors do not seem to be aware of this line of work that already separates the neural network learning power from any linear learners.

(2). Assuming that the training can minimize the objective function up to global optimal. While it is true that the neural network can fit all the training data in practice, it is very unclear whether they can actually find the global optimal of the regularized objective. This is quite an unrealistic assumption. The authors should at least extend their results to the case when the objective is <= some value, instead of only talking about the global optimal.


(3). Parallel architecture. The parallel network considered in this paper has all the intermediate layers being completely disjoint. This version of the parallel network is not what is used in practice. For example in ResNext, the parallel is layer-wise. The authors should at least clarify this difference instead of trying to present the result like "we consider the parallel networks as used in practice"




**Award:**

No

---

### Decision · Program_Chairs · 2022-09-14

Reject